# Fueling the Future: A Comprehensive Analysis and Forecast of Fuel Consumption Trends in U.S. Electricity Generation

**Md Monjur Hossain Bhuiyan [1,*]**, **Ahmed Nazmus Sakib [1]**, **Syed Ishmam Alawee [2]** and **Talayeh Razzaghi [3]**

[1] School of Aerospace and Mechanical Engineering, University of Oklahoma, Norman, OK 73019, USA; nazmus.sakib@ou.edu
[2] School of Data Science and Analytics, University of Oklahoma, Norman, OK 73019, USA; syed.ishmam@outlook.com
[3] School of Industrial and Systems Engineering, University of Oklahoma, Norman, OK 73019, USA; talayeh.razzaghi@ou.edu
[*] Correspondence: mhbhuiyan@ou.edu

**Abstract:** The U.S. Energy Information Administration (EIA) provides crucial data on monthly and annual fuel consumption for electricity generation. These data cover significant fuels, such as coal, petroleum liquids, petroleum coke, and natural gas. Fuel consumption patterns are highly dynamic and influenced by diverse factors. Understanding these fluctuations is essential for effective energy planning and decision making. This study outlines a comprehensive analysis of fuel consumption trends in electricity generation. Utilizing advanced statistical methods, including time series analysis and autocorrelation, our objective is to uncover intricate patterns and dependencies within the data. This paper aims to forecast fuel consumption trends for electricity generation using data from 2015 to 2022. Several time series forecasting models, including all four benchmark methods (Mean, Naïve, Drift, and seasonal Naïve), Seasonal and Trend Decomposition using Loess (STL), exponential smoothing (ETS), and the Autoregressive Integrated Moving Average (ARIMA) method, have been applied. The best-performing models are determined based on Root Mean Squared Error (RMSE) values. For natural gas (NG) consumption, the ETS model achieves the lowest RMSE of 20,687.46. STL demonstrates the best performance for coal consumption with an RMSE of 5936.203. The seasonal Naïve (SNaïve) model outperforms the others for petroleum coke forecasting, yielding an RMSE of 99.49. Surprisingly, the Mean method has the lowest RMSE of 287.34 for petroleum liquids, but the ARIMA model is reliable for its ability to capture complex patterns. Residual plots are analyzed to assess the models' performance against statistical parameters. Accurate fuel consumption forecasting is very important for effective energy planning and policymaking. The findings from this study will help policymakers strategically allocate resources, plan infrastructure development, and support economic growth.

**Keywords:** fuel consumption; forecasting; time series analysis; sustainable energy policies; RMSE

## 1. Introduction

The United States relies on a diverse array of energy sources, broadly categorized into primary sources, including fossil fuels, nuclear energy, and renewables, and secondary sources, represented by electricity generated from primary sources [1]. Measurement units vary across energy types, with liquid fuels being quantified in barrels or gallons, natural gas being quantified in cubic feet, coal in short tons, and electricity in kilowatts and kilowatt-hours [2]. The British thermal unit (Btu) serves as a standard for energy comparison, revealing that the total U.S. primary energy consumption reached 100.41 quadrillion Btu in 2022 [3,4]. In the realm of electricity generation, the United States employs a spectrum of sources and technologies that have evolved over time. The three principal categories encompass fossil fuels (coal, natural gas, and petroleum), nuclear energy, and renewables. Predominantly, steam turbines, drawn from fossil fuels, nuclear

energy, biomass, geothermal energy, and solar thermal energy, stand as the dominant force in electricity generation [5].

Other technologies contributing to this landscape include gas turbines, hydro turbines, wind turbines, and solar photovoltaics [6]. Notably, within the domain of fossil fuels, natural gas emerges as the predominant player, contributing approximately 40% to the electricity generated in the U.S. in 2022 [7]. Coal, occupying the third-largest share at around 18%, predominantly fuels steam turbines, with certain facilities converting coal to gas for utilization in gas turbines. Petroleum's contribution is nominal, constituting less than 1%, where residual fuel oil and petroleum coke are applied in steam turbines, and distillate fuel oil powers diesel engine generators [7]. Nuclear energy commands a substantial portion, accounting for nearly one-fifth of U.S. electricity, and it is generated through steam turbines and nuclear fission. Renewable energy sources, constituting approximately 22% of the total electricity generated in the U.S. in 2022, have witnessed substantial growth since 1990, when their contribution to utility-scale electricity generation was approximately 12% [8]. This discernible shift underscores the escalating significance of renewables in shaping the U.S. energy landscape [3,7]. In-depth analyses and insights from reputable sources further verify these trends and dynamics.

Understanding the fluctuations in fuel usage for generating electricity, as described by the U.S. Energy Information Administration, is quite difficult. The irregular nature of these consumption patterns highlights a critical need for precise predictive models, as their absence delays informed decision making, leaving us struggling with the dynamic shifts in energy demands. This urgent situation makes it important to start researching and developing better ways to predict future electricity needs. These improved forecasting techniques are crucial for planning ahead in the field of electricity generation [9]. Accurate predictions of fuel consumption not only facilitate efficient resource allocation and guide infrastructure development but also serve as the bedrock for informed energy policies [10]. Apart from just numbers and predictions, these forecasts have a large effect on decisions about where to invest money and how to plan for the environment, manage power grids, save costs, and come up with strategies for the market [11]. They emerge as key tools for emergency preparedness and play a pivotal role in planning a seamless transition to renewable energy, ultimately forging a path toward a stable, sustainable, and resilient energy sector [12]. As we investigate the statistics of this fluctuation, the intricate jump of fuel consumption in electricity generation unveils itself, highlighting the pressing need for foresight in steering the future of our energy landscape [10].

This study mainly focuses on constructing a robust forecasting model for fuel consumption in U.S. electricity generation, utilizing comprehensive data spanning the years 2015 to 2022. The method involves a systematic analysis of seasonal and trend patterns, a rigorous autocorrelation analysis, and refinement by eliminating biases from historical trends. Leveraging actual consumption data, the approach includes model validation and fine-tuning through a detailed comparison of forecasted and real consumption figures. By employing data from the U.S. Energy Information Administration, this eight-year analysis aims to explain the primary trend of fuel consumption for electricity generation across the entire United States, mitigating potential uncertainties associated with regional variations. The forecasting process integrates diverse benchmark methods, including STL, ETS, and ARIMA, facilitating a robust comparison to identify the most accurate model. Evaluation metrics, such as Mean Error (ME), Mean Percentage Error (MPE), Mean Absolute Error (MAE), Root Mean Squared Error (RMSE), and Mean Absolute Percentage Error (MAPE), guide the selection of the optimal model. Once identified, this model will be deployed to predict future fuel consumption trends, offering valuable insights for strategic energy planning. The combination of a comprehensive approach, diverse forecasting models, a specific fuel analysis, and a focus on policy impact distinguishes this work from previous studies in the field of fuel consumption forecasting for electricity generation. This research is crucial for informing stakeholders and decision makers, enabling them to make well-informed decisions on resource allocation, energy planning, and sector-specific strategies, with the

main goal of contributing to the formulation of sustainable energy policies for a stable and reliable national energy supply [13–15].

## 2. Analytical Framework

### 2.1. Data Source and Collection

The U.S. Energy Information Administration (EIA) serves as the primary data source for this study, offering crucial insights into monthly and annual fuel consumption for electricity generation. The comprehensive dataset provided by the EIA encompasses significant fuels that are essential for electricity generation, including coal, petroleum liquids, petroleum coke, and natural gas. The data utilized in this study were obtained from the U.S. Energy Information Administration and are accessible through their official website at www.eia.gov/electricity/monthly/ (accessed on 26 February 2024) [7,16]. The availability of monthly data spanning from 2015 to 2024 allows for a robust analysis of fuel consumption trends over an eight-year period. The dataset includes monthly data for coal, petroleum liquids, petroleum coke, and natural gas consumption specifically tailored for electricity generation purposes. The units of measurement for each fuel type are as follows:

- Coal: Thousand Tons;
- Petroleum Liquids: Thousand Barrels;
- Petroleum Coke: Thousand Tons;
- Natural Gas: Million Cubic Feet.

By incorporating data from the U.S. Energy Information Administration, this study conducts a comprehensive analysis of fuel consumption trends in electricity generation across the entire United States. The inclusion of monthly data from 2015 to 2024 facilitates the exploration of primary trends while mitigating potential uncertainties associated with regional variations. This transparent approach to data collection provides a solid foundation for the subsequent analysis and forecasting efforts outlined in this project proposal.

### 2.2. Statistical Methods

The analysis includes a detailed comparison of forecasting models based on statistical errors, such as Mean Error (ME), Root Mean Square Error (RMSE), Mean Absolute Error (MAE), Mean Percentage Error (MPE), Mean Absolute Percentage Error (MAPE), and Autocorrelation Function at lag 1 (ACF1). The significance of these error metrics is twofold: they quantify the accuracy and reliability of the forecasts and provide a statistical basis for model selection. Lower values of RMSE and MAPE, for example, indicate higher forecast accuracy. We further elaborate on the statistical underpinnings of these metrics, explaining how they are calculated and their relevance in assessing model performance. This error analysis is crucial for understanding why certain models outperform others and how the confidence intervals derived from these errors inform the precision of our forecasts.

### 2.3. Forecasting Models

In the methodology section, we employed a variety of time series forecasting models to analyze fuel consumption trends, including the Mean, Naïve, Drift, Seasonal Naïve, STL, ETS, and ARIMA methods. Each model was chosen for its ability to capture different aspects of the data's temporal dynamics. For instance, the Mean Model provides a basic benchmark by averaging past observations, while the Naïve method assumes the last observed value is the next period's value, which is useful for highly stable series. The Drift method extends the Naïve approach by allowing trends over time, and the Seasonal Naïve method accounts for seasonal patterns. Advanced models, like STL, ETS, and ARIMA, were selected for their flexibility in modeling complex patterns involving trends and seasonality. The best-performing model was determined based on the lowest Root Mean Square Error (RMSE), ensuring the most accurate forecasts. We applied these models to the dataset, systematically comparing their performance to identify the most accurate forecasting approach for different fuel types.

## 3. Data Analysis

### 3.1. Overall Trend Analysis of Fuel Consumption

The trends observed in fuel consumption over time, depicted in Figure 1, show the dynamics of various fuel types utilized for U.S. electricity generation. Natural gas consumption (Figure 1a) exhibits a consistent upward trajectory from January 2016 to January 2022, with occasional fluctuations [7,16]. This significant increase, supported by statistical evidence [16], is attributed to several factors, including the growing preference for natural gas as a cleaner alternative, increased efficiency in gas-powered plants, and a shift towards renewables in the energy mix. In contrast, coal consumption (Figure 1b) displays a fluctuating trend with distinct periods of growth and decline, reaching a significant trough in January 2020.

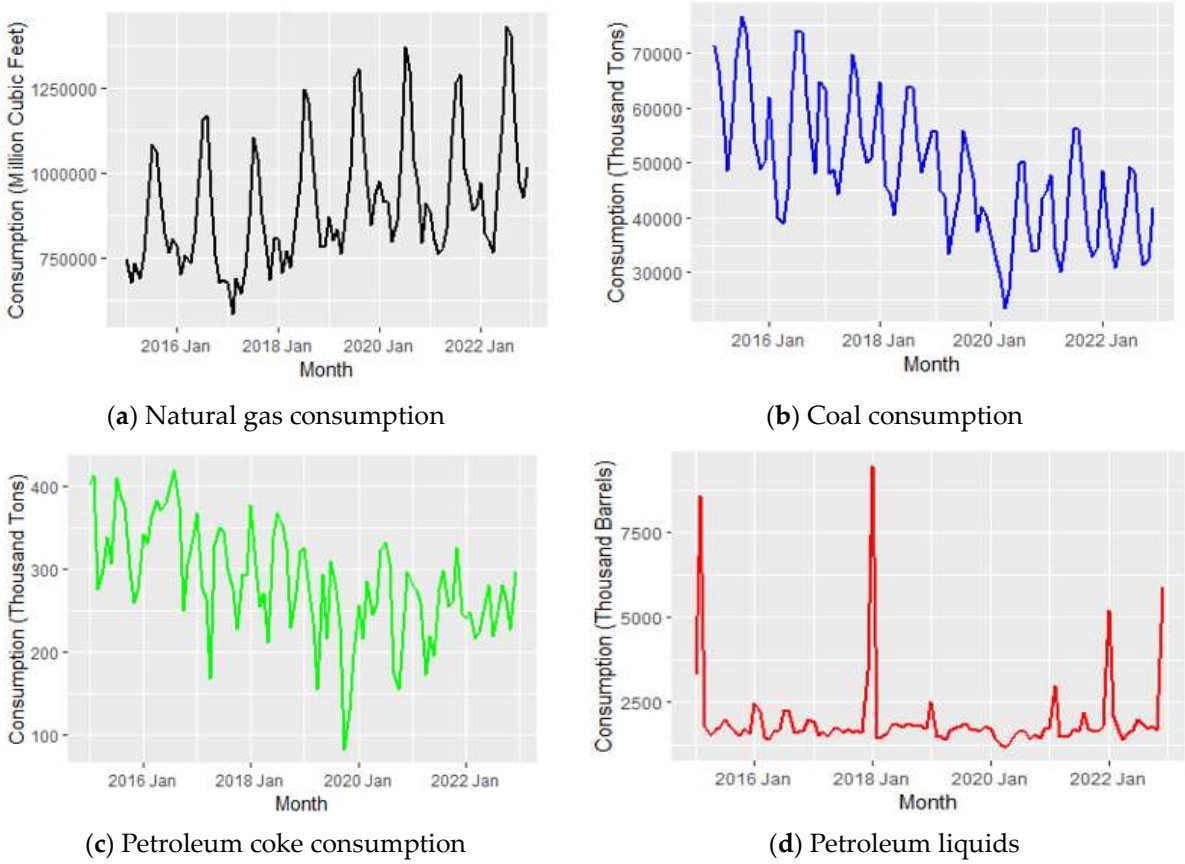

(**a**) Natural gas consumption

(**b**) Coal consumption

(**c**) Petroleum coke consumption

(**d**) Petroleum liquids

**Figure 1.** Trend analysis of different fuels' consumption for U.S. electricity generation over time.

This decline aligns with a broader global trend of decreasing reliance on coal, driven by environmental concerns, regulatory shifts, and the pursuit of cleaner energy sources [17]. Notably, the drop in January 2020 coincides with the onset of the COVID-19 pandemic, which induced a temporary reduction in industrial activities and energy demand, contributing to the observed drop in coal consumption. Similarly, petroleum coke (Figure 1c) usage exhibits variability, experiencing peaks at specific times. The visible drop in January 2020, concurrent with the pandemic's onset, can be attributed to a combination of reduced industrial activities, altered production patterns, and an overall decline in energy demand during the initial phases of the pandemic [18]. Petroleum liquids (Figure 1d) demonstrate a diverse trend with notable spikes, particularly in early 2018. This increase can be linked to a confluence of factors, including economic conditions, geopolitical events, and shifts in energy policies [19]. For instance, the spike in early 2018 may be associated with increased demand driven by economic growth and geopolitical factors affecting oil prices [20].

In summary, the observed trends emphasize the comprehensive nature of fuel consumption patterns, influenced by a complex interplay of economic, environmental, and technological factors. These notable changes, especially during the COVID-19 pandemic, highlight the importance of considering external influences when analyzing fuel consumption dynamics for strategic energy planning and policy formulation [21].

### 3.2. Seasonality Analysis of Fuel Consumption

Conducting a seasonality analysis of fuel consumption is vital for understanding recurring patterns and cycles in fuel consumption over time. It enables accurate forecasting and prediction, aiding in operational planning and resource management. The seasonality analysis of the consumption of different fuels is explained below.

#### 3.2.1. Seasonality Analysis of NG Consumption

In Figure 2a, the upward trajectory of NG consumption from June to August each year reflects a consistent seasonal pattern, particularly during the summer months. This observed peak aligns with that of Figure 2b, illustrating the typical seasonal demand for electricity, which is notably driven by increased air conditioning needs in warmer weather. The surge in NG consumption during these summer periods can be attributed to its crucial role in meeting the heightened electricity demand, with natural gas power plants playing a pivotal role. This seasonality underscores the importance of natural gas in addressing the specific requirements of peak periods, providing valuable insights into its usage dynamics. The ascending trend in NG consumption from 2016 to 2022 further emphasizes its increasing significance as an energy source for electricity generation, supported by advancements in natural gas technologies, its environmental advantages, and evolving energy policies.

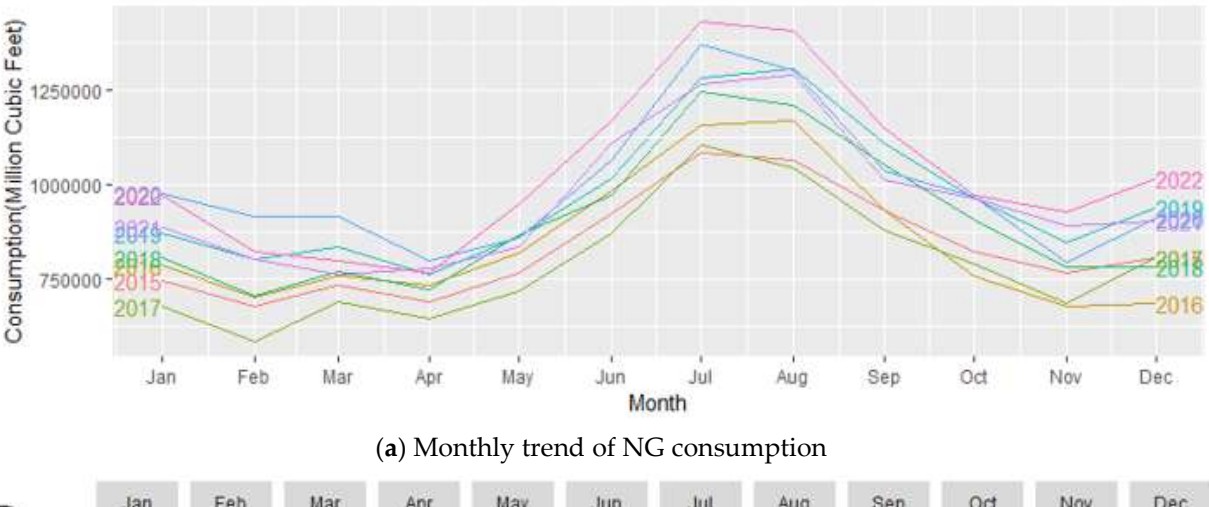

(**a**) Monthly trend of NG consumption

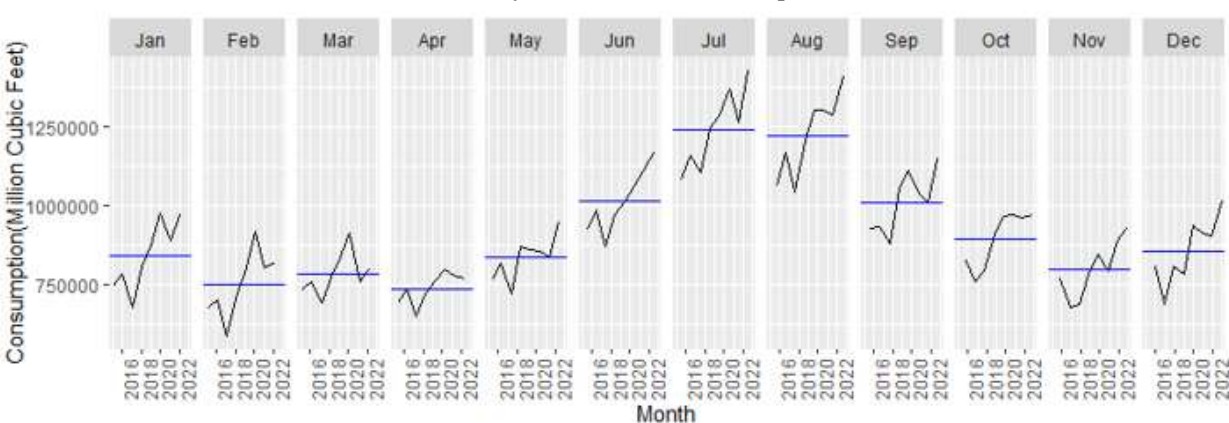

(**b**) Seasonality analysis of NG consumption

**Figure 2.** Monthly consumption and seasonality analysis of NG from 2015 to 2022.

### 3.2.2. Seasonality Analysis of Coal Consumption

The monthly trend and seasonality pattern observed in coal consumption from 2015 to 2022, shown in Figure 3, reveals a distinct seasonality pattern. The monthly trend, depicted in Figure 3a, illustrates a gradual decline in consumption from January to April, followed by an increase starting in April, reaching a peak in consumption during the summer months of July and August. It then gradually starts to decline through September and October. Subsequently, consumption reaches a relatively low level in November and experiences an upturn again in December. The seasonal analysis pattern, illustrated in Figure 3b, further reinforces this observed trend. It clearly shows the peak consumption during the summer months, indicating a significant surge in activity during this period. This repeating pattern shows that coal usage is strongly influenced by the seasons. Coal consumption increases during the summer when there is a higher demand for cooling, and then it decreases again as the winter becomes closer.

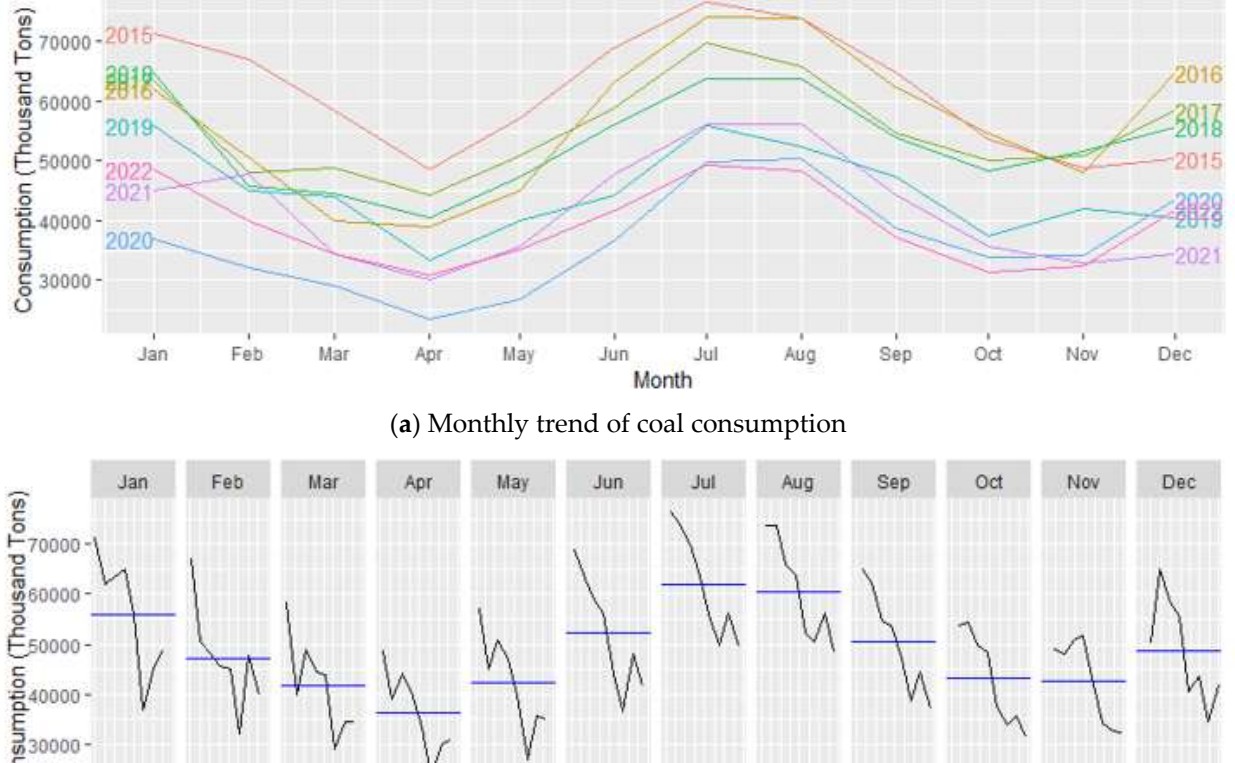

(**a**) Monthly trend of coal consumption

(**b**) Seasonality analysis of coal consumption

**Figure 3.** Monthly consumption and seasonality analysis of coal from 2015 to 2022.

The numbers behind this trend show that there is a regular change in coal usage each month. By looking at the differences between the highest and lowest points, we can see how much seasonal changes affect coal consumption [16]. The observed cyclicality aligns with established industry knowledge on the impact of weather-related demand fluctuations on energy consumption patterns [22,23]. This seasonality analysis, presented in Figure 3a,b, contributes valuable insights for energy planners and policymakers, enabling informed decisions to address the cyclical nature of coal consumption in the context of changing climate and energy demand dynamics.

### 3.2.3. Seasonality Analysis of Petroleum Coke Consumption

The monthly trend of petroleum coke consumption from 2015 to 2022, depicted in Figure 4a, exhibits noteworthy fluctuations, revealing distinctive patterns across different months. The seasonality analysis in Figure 4b further explains these trends, highlighting recurring patterns within the specified timeframe. In particular, petroleum coke consistently reaches its lowest consumption levels in October, with a recurring trend of reduced usage also observed in November. Conversely, the summer months of July and August consistently exhibit the highest consumption levels, which is indicative of a seasonal peak in fuel usage. This observed seasonality aligns with industry practices, where heightened energy demands during the summer, which are possibly attributed to increased industrial activities and higher temperatures, drive an uptick in petroleum coke consumption. The recurrent nature of these patterns emphasizes the importance of considering seasonal variations for accurate forecasting and strategic energy planning. These observations are grounded in data from reputable sources, providing a robust foundation for understanding the intricate seasonality dynamics in petroleum coke consumption.

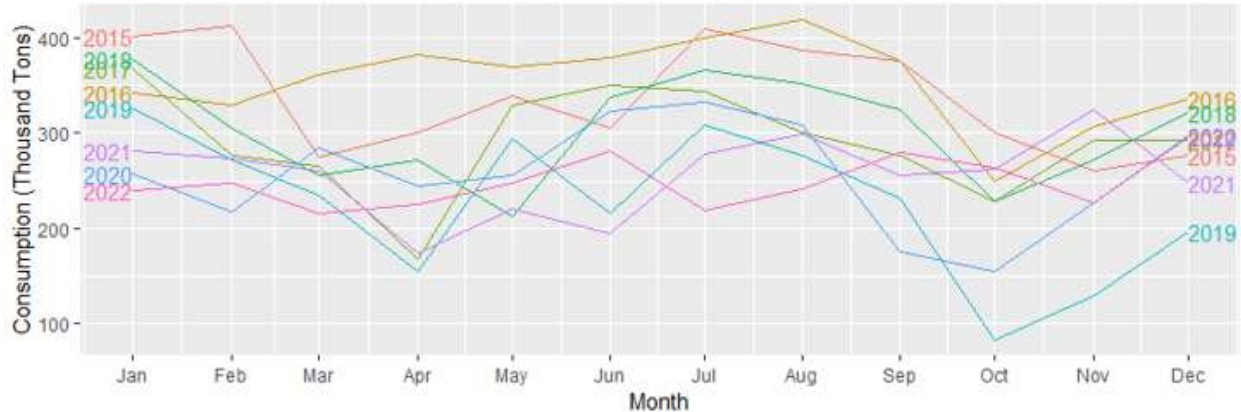

(**a**) Monthly trend of petroleum coke consumption

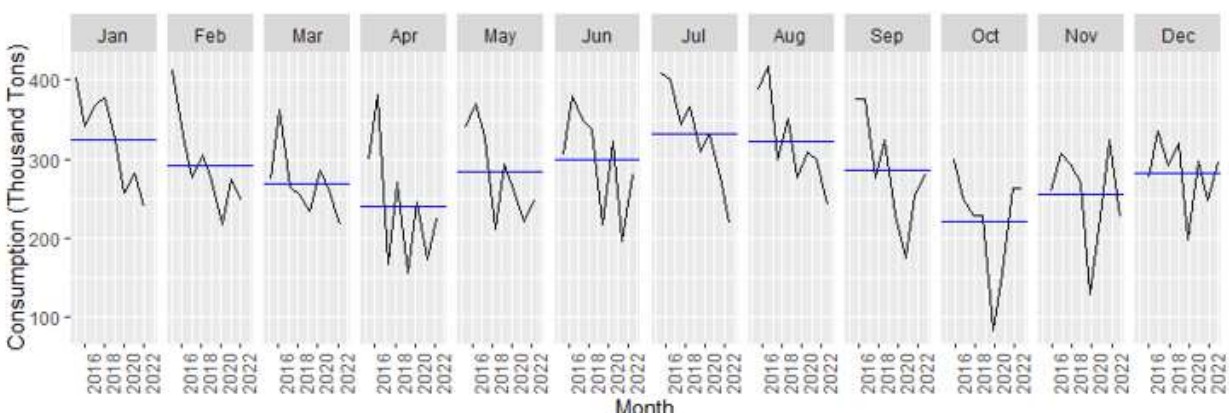

(**b**) Seasonality analysis of petroleum coke consumption

**Figure 4.** Monthly consumption and seasonality analysis of petroleum coke from 2015 to 2022.

### 3.2.4. Seasonality Analysis of Petroleum Liquid Consumption

The graphical representations in Figure 5a,b offer comprehensive insights into the monthly trends and seasonality analysis of petroleum liquid consumption in the United States from 2015 to 2022. In Figure 5a, the monthly trend reveals notable fluctuations, bringing attention to abnormal spikes in February 2015, January 2018, January 2022, and December 2022. These anomalies may be attributed to various factors, such as geopolitical events influencing global oil markets, economic shifts impacting demand, or specific regula-

tory changes affecting petroleum consumption. Figure 5b, showing the seasonality analysis, provides a deeper understanding of recurring patterns. The observed regular fluctuations suggest that seasonal changes, economic cycles, or global events have a potential influence on petroleum liquid consumption. For instance, heightened demand during winter months or economic upturns may contribute to periodic spikes. These insights underscore the need for a nuanced understanding of the complex factors influencing petroleum liquid consumption, integrating considerations beyond mere temporal patterns. The observed data from the U.S. Energy Information Administration (EIA) provide additional context for these observations [7,22,23].

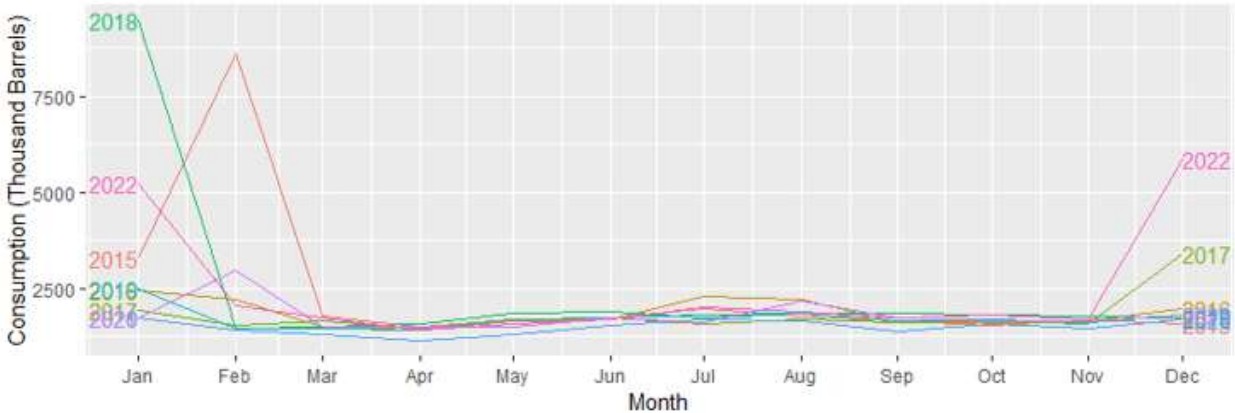

(**a**) Monthly trend of petroleum liquid consumption

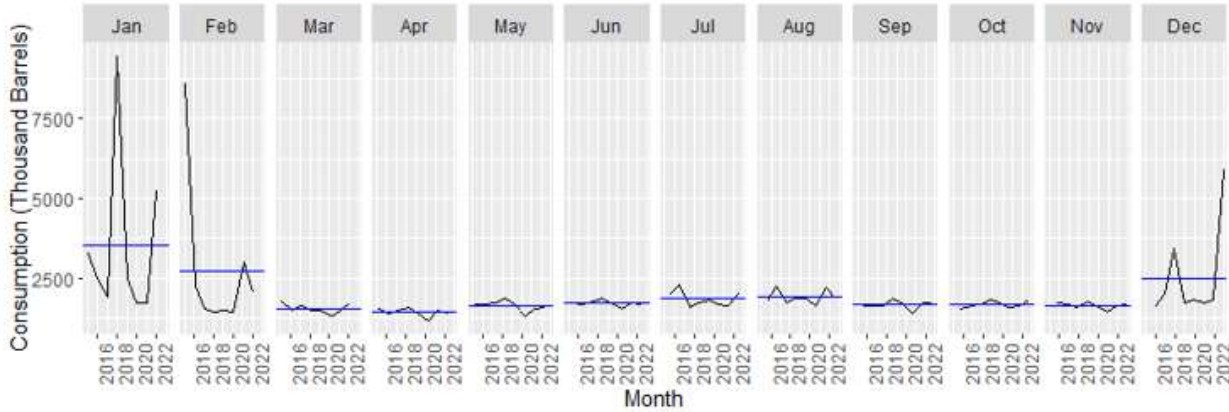

(**b**) Seasonality analysis of petroleum liquid consumption

**Figure 5.** Monthly consumption and seasonality analysis of petroleum liquids from 2015 to 2022.

*3.3. ACF and PACF Analyses of Consumption of Different Fuels*

Analyzing the Autocorrelation Function (ACF) and Partial Autocorrelation Function (PACF) plots is a pivotal step in formulating an accurate forecasting model for fuel consumption in electricity generation spanning the years 2015 to 2022. These plots play crucial roles in detecting seasonality and identifying autoregressive (AR) and moving average (MA) components, which are vital for comprehending how fuel consumption changes over time. Autoregressive (AR) and moving average (MA) components are vital in a time series analysis; they are identified through Autocorrelation Function (ACF) and Partial Autocorrelation Function (PACF) plots. The AR model order is determined by ACF plot lags where autocorrelation drops, while the MA model order is identified by PACF plot lags where partial autocorrelation diminishes. ARIMA models integrate AR and MA components, addressing non-stationarity for comprehensive time series forecasting. The ACF plot provides a comprehensive overview of autocorrelation at various lags, offering insights

into longer-term patterns such as seasonality in fuel usage during specific calendar periods. This relationship can be expressed mathematically as shown in Equation (1):

$$\text{ACF (k)} = \frac{\sum_{t=k+1}^{n} \left( X_t \ - \hat{X} \right) \left( X_{t-k} \ - \hat{X} \right)}{\sum_{t=1}^{n} \left( X_t \ - \hat{X} \right)^{2}} \tag{1}$$

where

- $k$ is the lag;
- n is the total number of observations in the time series;
- $X_t$ is the value of the time series at time t;
- X is the mean of the time series.

Simultaneously, the PACF plot focuses on the direct relationship between the current observation and observations at individual lags, aiding in model selection by revealing correlations within the fuel consumption time series. This approach allows for a nuanced understanding of fuel consumption patterns, capturing both broader trends and specific lagged relationships. This relationship can be expressed mathematically as shown in Equation (2) [24,25].

$$\text{PACF}_k = \text{corr} \left( L_t, L_{t-k} \,|\, L_{t-1}, \ldots\ldots\ldots L_{t-k-1} \right) \tag{2}$$

Utilizing the ACF enables the identification of overarching temporal patterns, while the PACF emphasizes the immediate connections within the fuel consumption data. By employing Equation (2), the PACF quantifies the correlation at lag, Lt, extending up to a defined number of lags, contributing to a more robust analysis [24,25]. The significance of ACF and PACF analyses lies in their ability to uncover temporal dependencies, aiding in the formulation of a forecasting model that accounts for seasonality and lagged relationships, ultimately enhancing the accuracy of the predictions. This methodological approach is fundamental for gaining a comprehensive understanding of the intricate fuel consumption patterns that are crucial for effective energy planning and policy formulation.

### 3.3.1. Autocorrelation Analysis of NG Consumption

In the specific analysis focused on natural gas consumption using a lag of 36 in the ACF plot, the majority of lines are observed to cross the blue dashed line on the positive side, suggesting a significant positive autocorrelation at lag 36 (Figure 6a). The positive crossings indicate a correlation between values at a given month and those from 36 months prior, revealing a prolonged cyclicality in consumption trends. As the line (blue dashed) typically represents the 95% confidence interval for the autocorrelation values, points outside this interval may indicate significant autocorrelation. This confidence interval is essential because it helps determine whether observed autocorrelation values are statistically significant or simply due to random variation. By setting the confidence level at 95%, we establish a threshold for significance, allowing us to identify meaningful autocorrelation patterns with a high degree of confidence. In addition, in the PACF plot shown in Figure 6b, the lines crossing the blue dashed line up to lag 15 indicate statistically significant partial autocorrelations, and partial autocorrelations beyond this point are not statistically significant.

Lag 36 is chosen for the analysis of natural gas consumption to capture recurring seasonal patterns or business cycles, aligning with standard practices in time series analysis. This lag allows for the exploration of relevant patterns while maintaining a balance between examining autocorrelation over an adequate timeframe and capturing significant temporal dynamics. By considering the confidence interval alongside lag selection, the autocorrelation function (ACF) analysis offers valuable insights into the structure of autocorrelation, facilitating the development of robust forecasting models and informed decision making.

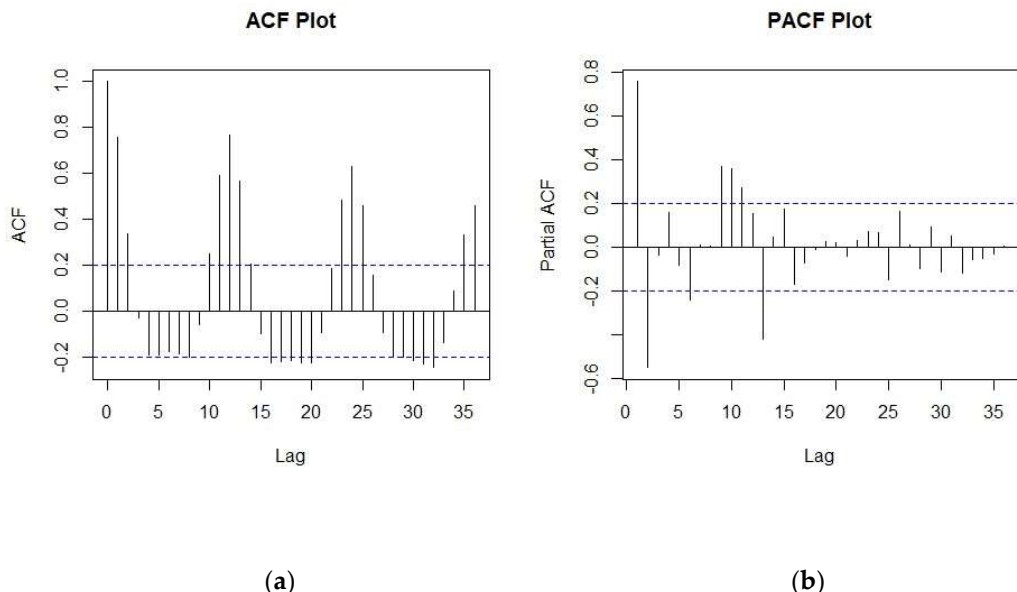

(**a**)  (**b**)

**Figure 6.** Autocorrelation analysis of NG consumption using ACF and PACF plots. (**a**) ACF plot, NG consumption; (**b**) PACF plot, NG consumption.

### 3.3.2. Autocorrelation Analysis of Coal Consumption

The ACF plot shown in Figure 7a for coal consumption with a lag of 36 provides insights into the autocorrelation structure, indicating the presence of seasonality within the first 27 months, followed by a change in the pattern. The positive crossing on the ACF plot up to lag 27 indicates significant positive autocorrelation at these lags. This suggests the presence of a repeating pattern or seasonality within the first 27 months of the coal consumption data. The crossing to the negative side at lag 27 could indicate a reversal in the autocorrelation pattern. Moreover, the PACF plot shown in Figure 7b indicates significant autocorrelation up to lag 13, suggesting strong dependencies within the data up to that point. However, beyond lag 13, the autocorrelation decreases as the lag increases. The decrease in the number of lines crossing the confidence interval as the lag increases may suggest a decline in the partial autocorrelation, indicating that the direct influence of observations on other observations diminishes as the lag increases.

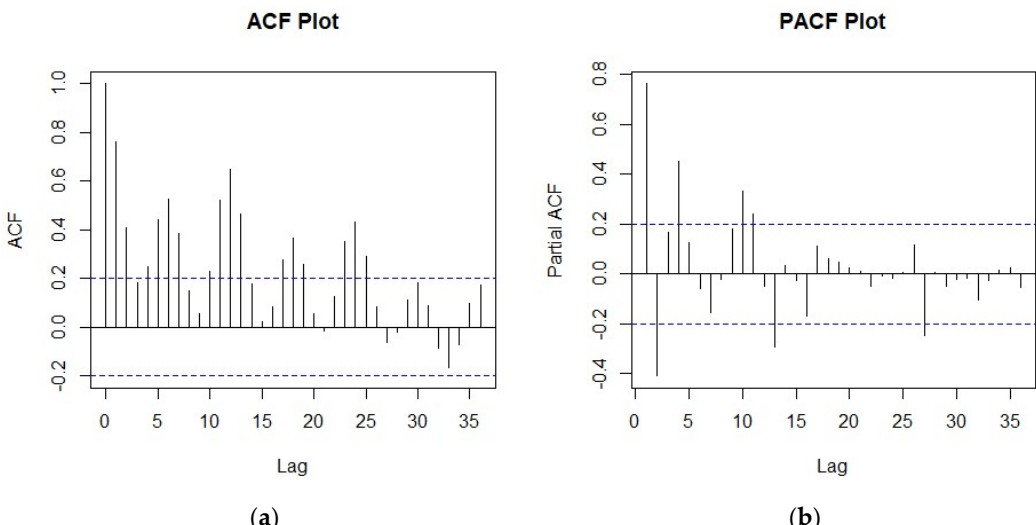

(**a**)  (**b**)

**Figure 7.** Autocorrelation analysis of coal consumption using ACF and PACF plots. (**a**) ACF plot, coal consumption; (**b**) PACF plot, coal consumption.

### 3.3.3. Autocorrelation Analysis of Petroleum Coke Consumption

The ACF plot for petroleum coke consumption, depicted in Figure 8a, indicates a positive correlation between observations at different lags, with the strength of correlation decreasing as the lag increases. The diminishing correlation suggests a changing influence of past observations on the current consumption pattern, possibly reflecting evolving trends or temporal dynamics. Conversely, the PACF plot (Figure 8b) does not show significant correlations, emphasizing the specific and direct influences captured by partial autocorrelations at each lag.

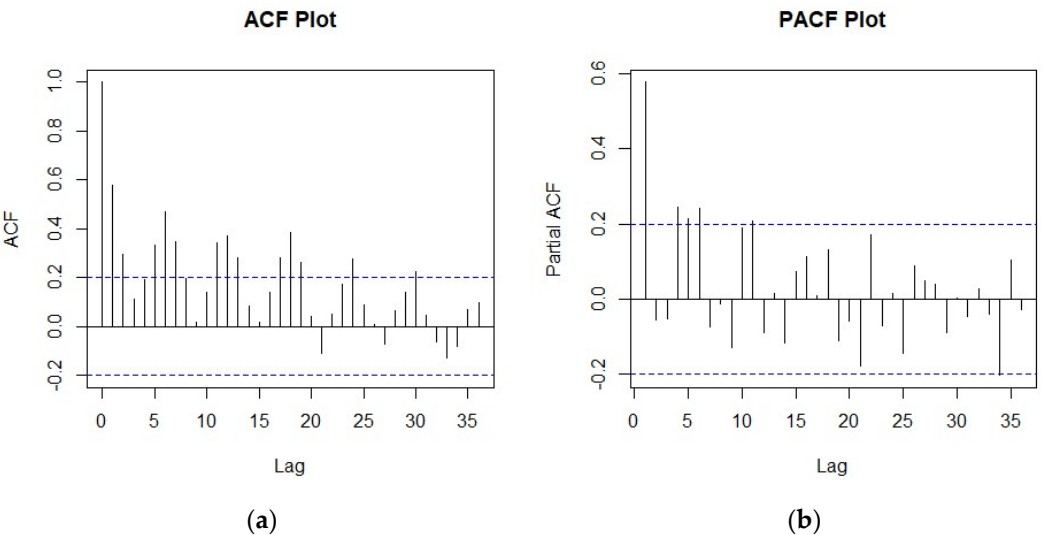

(**a**)　　　　　　　　　　　　　　　(**b**)

**Figure 8.** Autocorrelation analysis of coal consumption using ACF and PACF plots. (**a**) ACF plot, petroleum coke consumption; (**b**) PACF plot, petroleum coke consumption.

### 3.3.4. Autocorrelation Analysis of Petroleum Liquid Consumption

In the ACF plot of petroleum liquid consumption, the absence of lines crossing the dashed line, except at lag 0 and lag 35, indicates a lack of significant autocorrelation at most lags (Figure 9a). At lag 0, the line crossing the blue dashed line suggests a correlation with itself, which is expected. The PACF plot in Figure 9b does not show significant correlations, emphasizing the specific and direct influences captured by partial autocorrelations at each lag.

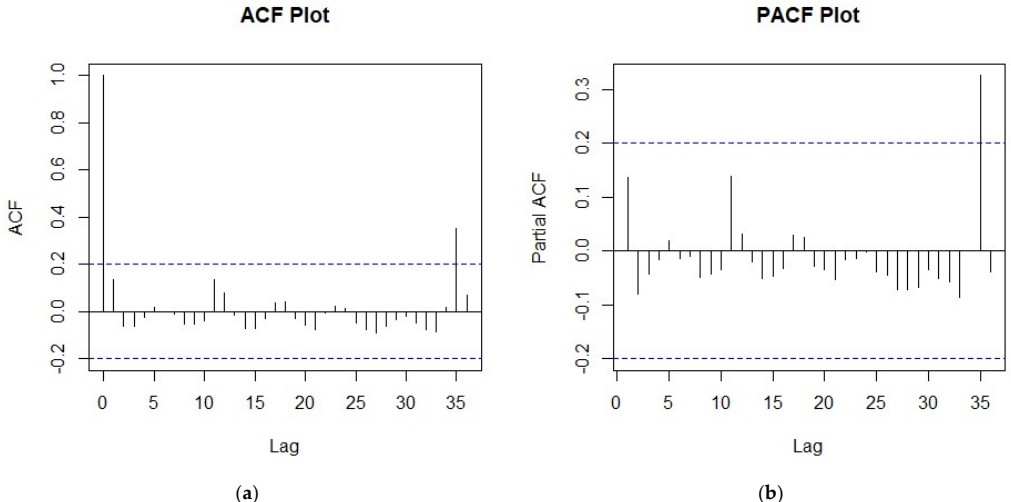

(**a**)　　　　　　　　　　　　　　　(**b**)

**Figure 9.** Autocorrelation analysis of petroleum liquid using ACF and PACF plots. (**a**) ACF plot, petroleum liquid consumption; (**b**) PACF plot, petroleum liquid consumption.

*3.4. Forecasting Models for Consumption of Different Fuels*

The development of forecasting models holds vital importance in anticipating and adapting to the dynamic demand profiles inherent in fuel consumption, particularly for natural gas (NG), for electricity generation [26]. In this research work, we deployed four distinct forecasting methodologies to project fuel consumption for electricity generation in the United States for the upcoming years 2023 and 2024. The models experienced a thorough training phase spanning an 8-year period from January 2015 to December 2022, followed by careful validation using test data from January 2023 to August 2023. To comprehensively assess their performance, we conducted a detailed comparative analysis, evaluating the forecasting models based on their error metrics. The effectiveness of each model was further analyzed through a precise comparison of forecasted data against actual consumption figures [12]. The applied models along with the mathematical formula can be described accordingly.

3.4.1. Benchmark Models

The benchmark methods serve as crucial reference points in fuel consumption forecasting, providing intuitive approaches for comparison with more sophisticated models [27,28]. Benchmark approaches constitute foundational methodologies in time series forecasting, characterized by their simplicity and practicality.

Mean Model

The Mean approach employs the average of historical observations to project future values [29]. The Mean Model, often used as a baseline for comparison, forecasts future values by calculating the average of all past observations. This simplistic approach is grounded in the assumption that the future will closely mirror the average historical behavior of the dataset, making no adjustments for any trends or seasonal patterns that may exist. The model's simplicity lies in its disregard for complex behaviors, focusing instead on the central tendency of the data. The mathematical formula can be written as shown in Equation (3).

$$\hat{Y}_t = \frac{1}{n}\sum_{i=1}^{n} y_i \tag{3}$$

where $\hat{Y}_t$ is the forecasted value for the future period, t; $n$ is the total number of observations; and $y_i$ represents each observed value.

Naïve Model

The Naïve Model provides forecasts by simply assuming the next value will be identical to the last observed value. Its effectiveness and simplicity make it a reliable benchmark for evaluating the performance of more sophisticated models, especially in datasets where trend and seasonality are minimal or absent. Equation (4) shows the mathematical formula as follows:

$$\hat{Y}_{t+1} = Y_t \tag{4}$$

Here, $\hat{Y}_{t+1}$ is the forecasted value for the next time period, $t + 1$, and $y_t$ is the observed value at time $t$.

Drift Model

The Drift Model extends the Naïve Model by incorporating a linear trend into the forecasts. This trend is calculated based on the average change observed in the historical data, making it suitable for datasets with a consistent trend over time. The mathematical formula is shown in Equation (5).

$$\hat{Y}_{t+h} = Y_t + h\left(\frac{Y_t - Y_1}{t - 1}\right) \tag{5}$$

where *h* represents the number of periods ahead for which the forecast is being made.

Seasonal Naïve Model

The Seasonal Naïve Model forecasts future values based on the last observed value from the same season in the previous cycle. It captures seasonality by assuming that patterns repeat at a fixed seasonal frequency. The mathematical formula can be written as follows (Equation (6)):

$$\hat{Y}_{t+m} = Y_{t+m-s} \tag{6}$$

where *m* indicates the forecasting interval and *s* indicates the length of the seasonal cycle.

### 3.4.2. STL (Seasonal and Trend Decomposition Using Loess)

Decomposition techniques constitute crucial elements in the nuanced analysis of time series data, facilitating the discernment and isolation of pivotal components that are essential for unveiling trends, seasonal fluctuations, and cyclical patterns. One such sophisticated method is STL, denoting Seasonal and Trend Decomposition using Loess. This method stands out for its robustness in handling outliers and its adaptability in addressing seasonal time series characterized by frequencies exceeding one. Unlike methodologies confined to specific temporal resolutions, such as monthly or quarterly intervals, STL demonstrates versatility by accommodating a broader spectrum of seasonal patterns, rendering it particularly well suited for precise fuel consumption forecasting [30,31]. The mathematical formula can be written as follows (Equation (7)):

$$\hat{Y}_t = T_t + S_t + R_t \tag{7}$$

where $Y_t$ is the original time series, $T_t$ is the trend component, $S_t$ the seasonal component, and $R_t$ is the remainder or residual.

### 3.4.3. ETS (Error, Trend, Seasonality)

The ETS (Error, Trend, Seasonality) forecasting methodology, renowned for its efficacy in time series prediction, is instrumental in modeling and anticipating fuel consumption patterns. This approach dissects time series data into three fundamental components: error, trend, and seasonality. The error term encapsulates stochastic fluctuations, the trend encapsulates long-term directional movements, and seasonality captures recurring patterns at fixed intervals. Diverse ETS model variations, such as ETS (AAA), ETS (AAN), and ETS (MAM), tailor to distinct time series characteristics by combining different error, trend, and seasonality components. Their adaptability to varied time series patterns and capacity to unveil insights into future trends make ETS models pervasive in forecasting applications. Equation (8) shows the mathematical formulation.

$$\hat{Y}_t = T_t \times S_t \times E_t \tag{8}$$

### 3.4.4. ARIMA (Autoregressive Integrated Moving Average)

ARIMA models, encompassing autoregressive, differencing, and moving average components, offer a comprehensive framework for forecasting time series data. These models are adept at handling data with underlying trends and autocorrelations, providing a robust method for predicting future values. They combine autoregressive (AR) terms, differencing (I) to achieve stationarity, and moving average (MA) terms, making them suitable for data with trends and seasonal patterns. The pivotal parameters of the ARIMA model encompass the autoregressive component (*p*), which is vital for discerning correlations with antecedent values; the integrated component (*d*) determining the order of differencing requisite for achieving stationarity; and the moving average component (*q*), which is effective in accounting for correlations with prior prediction errors. The optimal values for '*p*', '*d*', and '*q*' are calculated through optimization techniques, such as grid search. After training on historical data, the ARIMA model proficiently extrapolates future

NG consumption values, adeptly accommodating both transient oscillations and enduring trends within the temporal dataset. This renders ARIMA an invaluable tool for precise and holistic forecasting within the realm of energy consumption [32].

$$ARIMA(p, d, q)$$

where $p$ is the order of the autoregressive part, $d$ is the degree of differencing needed to achieve stationarity, and $q$ is the order of the moving average part.

Each of these models contributes uniquely to the forecasting process, enabling a nuanced understanding of the time series data's behavior. By applying a combination of these models, this study leverages a broad spectrum of statistical tools to predict fuel consumption trends with greater accuracy and reliability.

3.4.5. An Analysis of the Consumption of Fuels Using Forecasted Methods
Forecasting Models for NG Consumption

Figure 10 serves as a visual representation of the forecasted NG consumption for the years 2023 and 2024, employing diverse forecasting methodologies. The benchmark methods, STL decomposition, ETS method, and ARIMA method each contribute to a multi-faceted analytical approach, offering an understanding of the varied strategies employed in forecasting. This rigorous methodology adheres to industry standards and contributes valuable insights to the realm of energy planning and policy formulation [33].

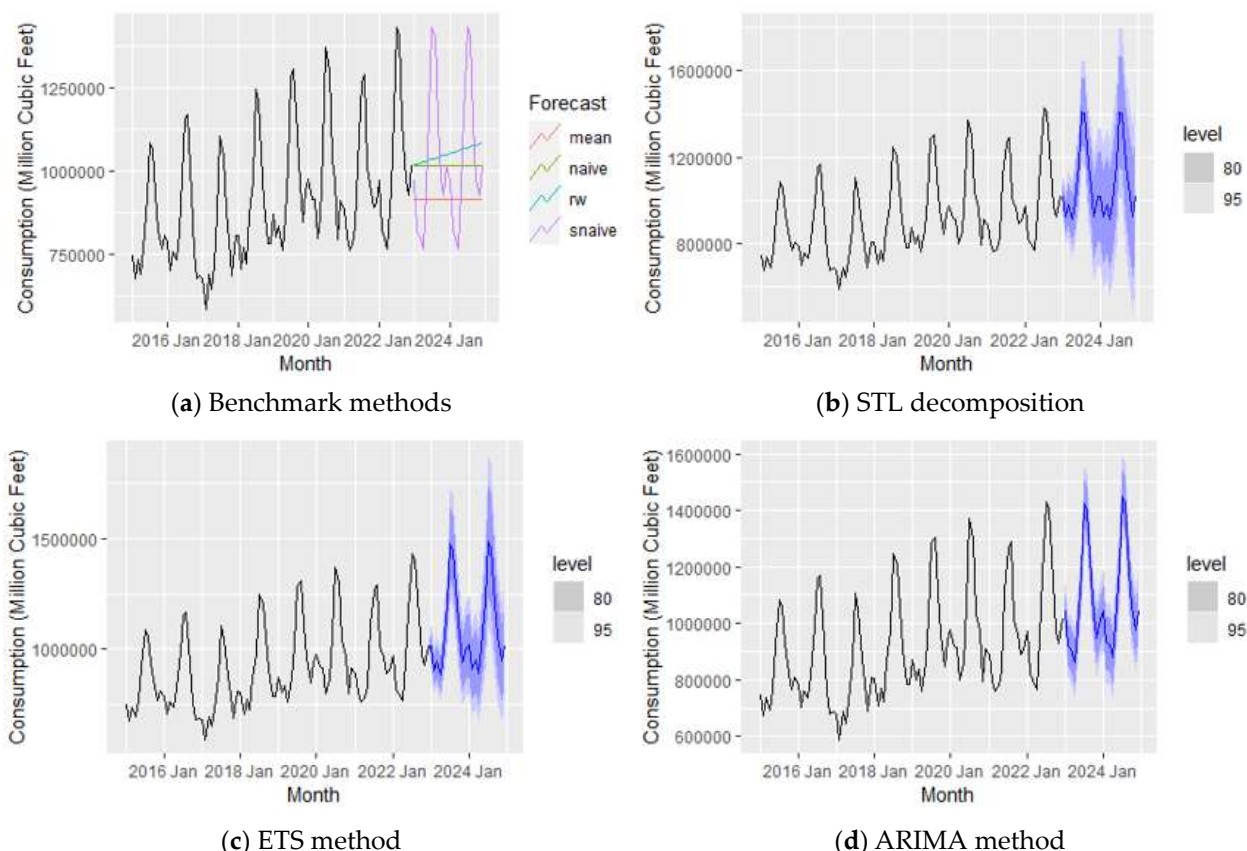

(**a**) Benchmark methods      (**b**) STL decomposition

(**c**) ETS method      (**d**) ARIMA method

**Figure 10.** Forecasting NG consumption for the years 2023 and 2024 using different forecasting methods.

Particularly, among the illustrated benchmark methods (refer to Figure 10a), the SNaïve technique distinguishes itself by adeptly capturing the majority of fluctuations in fuel consumption. The forecasting outcome following STL decomposition is illustrated in Figure 10b, attesting to its efficacy in enhancing predictive accuracy.

Figure 10c illustrates the forecasted natural gas (NG) consumption data using the ETS method. The ETS decomposition plot in Figure 11, derived from the ETS (M, N, M) model—signifying multiplicative error, no trend, and multiplicative seasonality—was selected based on the minimal values of AIC, AICc, and BIC, which were determined through the ETS () function in R. The estimated parameters for exponential smoothing include $\alpha = 0.583$ and $\Upsilon = 0.0001$, with a calculated $\sigma 2$ of 0.0024. Notably, $\alpha$ at 0.583 denotes a moderate emphasis on recent observations in forecasting, and the lower $\sigma 2$ value suggests heightened stability and predictability. These parameters assume a pivotal role in governing the rate of change for error, trend, and seasonality components, offering flexibility through $\alpha$ and $\Upsilon$ in adjusting level and trend, respectively.

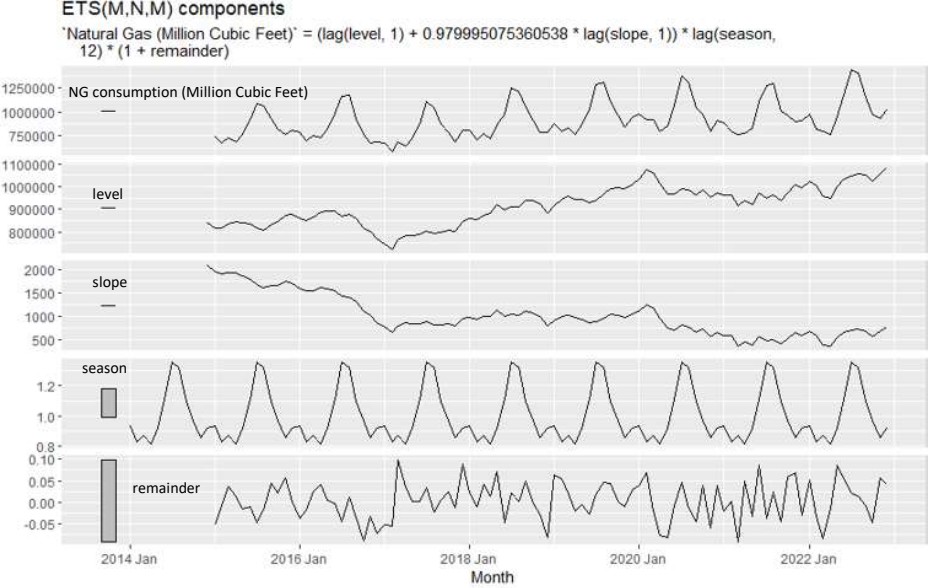

**Figure 11.** ETS (M, N, M) decomposition plot of NG consumption.

The forecasted natural gas (NG) consumption for electricity generation in 2023 and 2024, derived from the ARIMA forecasting method, is visually presented in Figure 10d, thereby showcasing the model's predictive prowess in delineating future trends.

Forecasting Models for Coal Consumption

In the prediction of coal consumption for electricity generation in the U.S., we applied various forecasting methods, including the benchmark method, STL decomposition, ETS method, and ARIMA method, as outlined in Section 5. The coal consumption values from 2015 to 2022 were used as input data for predicting the values for the years 2023 and 2024. Figure 12a–d visually represent the forecasted coal consumption values obtained through different methods.

Forecasting Models for Petroleum Coke Consumption

The forecasting process for petroleum coke consumption in the years 2023 and 2024 followed the same methodology discussed in Section 5. Similar forecasting methods were applied to determine the most effective approach. Figure 13 visually presents the forecasted values for petroleum coke consumption obtained through the benchmark, STL decomposition, ETS, and ARIMA methods.

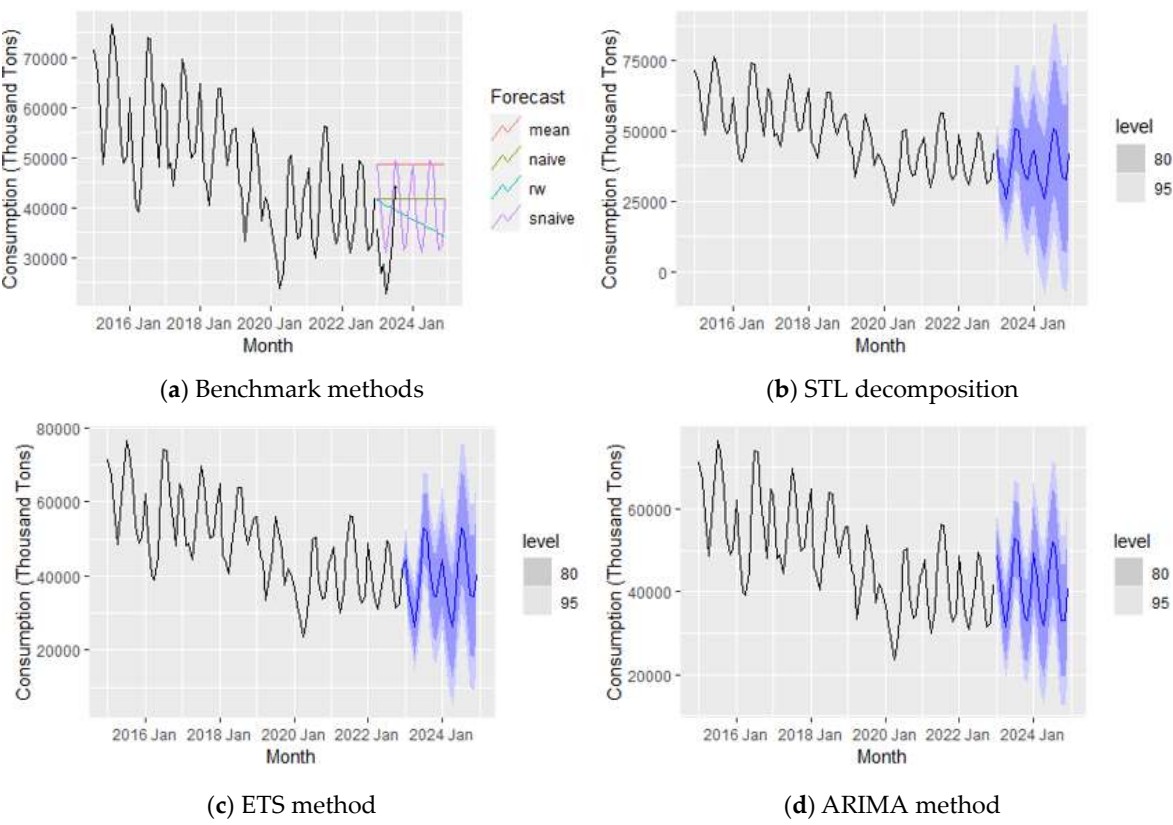

(**a**) Benchmark methods        (**b**) STL decomposition

(**c**) ETS method        (**d**) ARIMA method

**Figure 12.** Forecasting coal consumption for the years 2023 and 2024 using different forecasting methods.

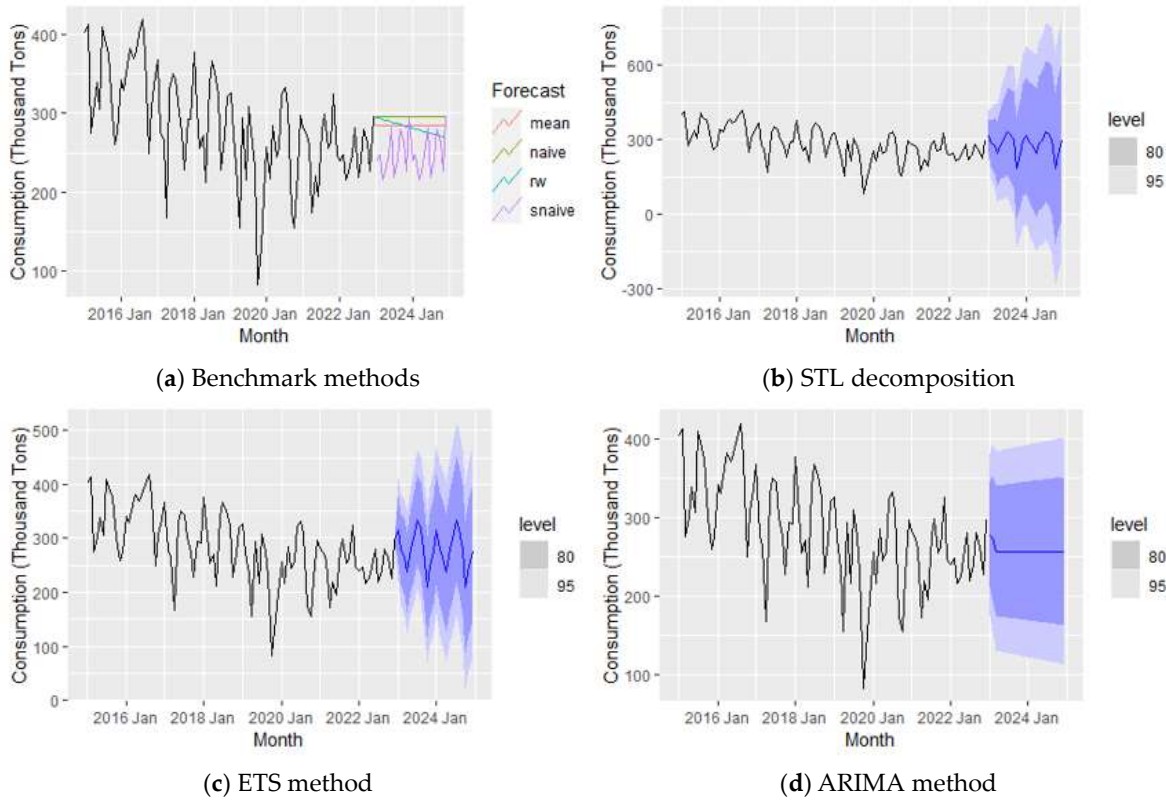

(**a**) Benchmark methods        (**b**) STL decomposition

(**c**) ETS method        (**d**) ARIMA method

**Figure 13.** Forecasting petroleum coke consumption for the years 2023 and 2024.

Forecasting Models for Petroleum Liquid Consumption

Created by applying forecasting methods similar to those discussed in Section 5, Figure 14 visually shows the forecasted values for petroleum liquid consumption obtained through the benchmark, STL decomposition, ETS, and ARIMA methods.

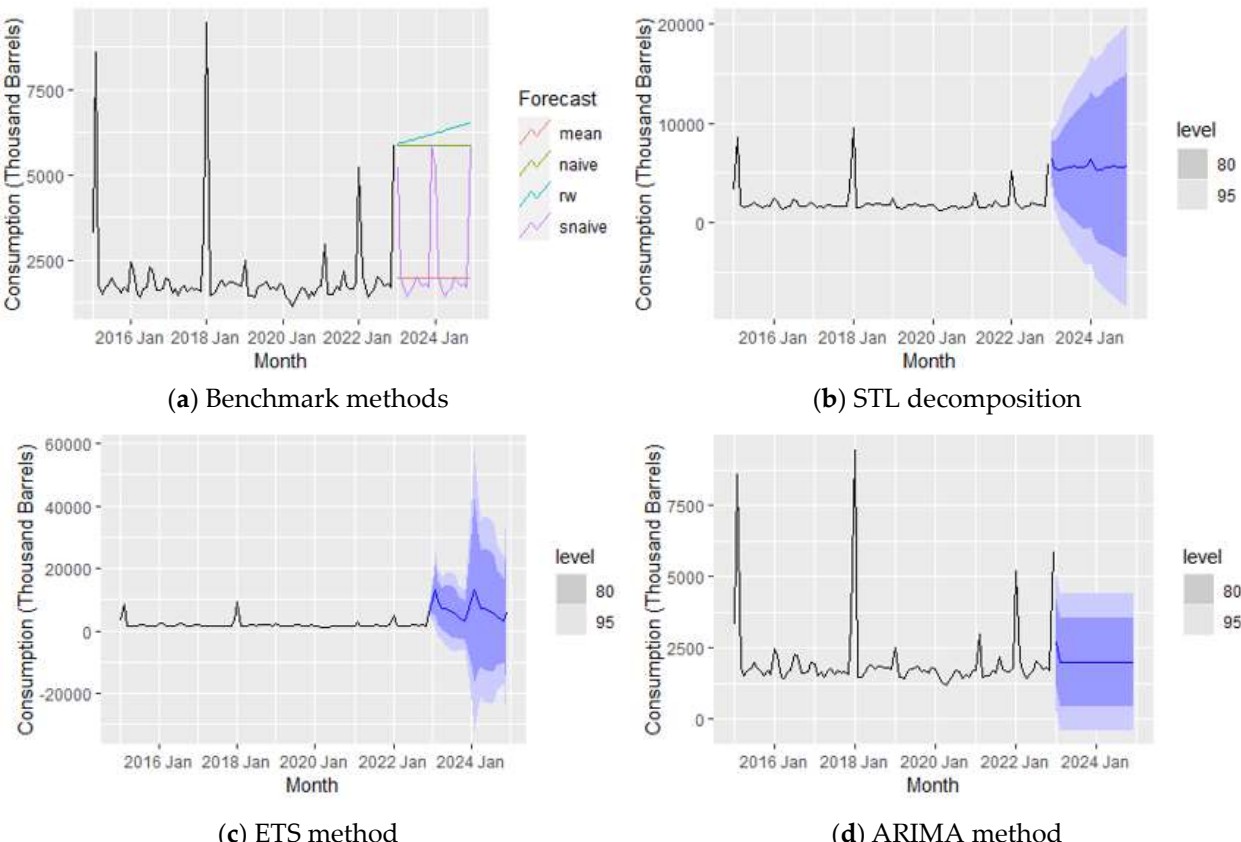

(**a**) Benchmark methods      (**b**) STL decomposition

(**c**) ETS method      (**d**) ARIMA method

**Figure 14.** Forecasting petroleum liquid consumption for the years 2023 and 2024.

*3.5. Model Comparison in Terms of Errors for Energy Streams*

The evaluation of the models' performance in forecasting energy streams is predicated on a robust statistical analysis of the forecast errors on both the training and testing datasets. The models selected for comparison encompass a range of statistical and machine learning approaches, each with distinct assumptions and complexities. To ensure a comprehensive and unbiased comparison, a suite of error metrics, each capturing different aspects of forecast accuracy and bias, were employed. Each model was applied to the same training and testing datasets to ensure accurate comparison. The error metrics used for comparison are standard in forecasting literature and include the Mean Error (ME), Root Mean Square Error (RMSE), Mean Absolute Error (MAE), Mean Percentage Error (MPE), Mean Absolute Percentage Error (MAPE), and the first autocorrelation of the forecast errors (ACF1). The ME is calculated as shown in Equation (9).

$$ME = \sqrt{\frac{1}{n}\sum_{t=1}^{n}\left(Y_t - \hat{Y}_t\right)} \tag{9}$$

It measures the average deviation of the forecasts ($\hat{y}_t$) from the actual values ($y_t$), indicating a bias if the ME is significantly different from zero. The RMSE is sensitive to

outliers and provides a measure of the dispersion of forecast errors. The RMSE is computed using Equation (10).

$$RMSE = \sqrt{\frac{1}{n}\sum_{t=1}^{n}(Y_t - \hat{Y}_t)^2} \tag{10}$$

This metric penalizes larger errors more severely and provides a measure of the magnitude of the error. The Mean Absolute Error (MAE) is similar to the RMSE in measuring the magnitude of errors but does not square the deviations, thus offering a linear scale of error. The MAE is given by the following equation:

$$MAE = \frac{1}{n}\sum_{t=1}^{n}|Y_t - \hat{Y}_t| \tag{11}$$

It represents the average magnitude of the errors in a set of forecasts without considering their directions. The Mean Percentage Error (MPE) and Mean Absolute Percentage Error (MAPE) are relative error metrics, providing a perspective of errors in terms of the observed data's magnitude. The MPE is calculated as follows (Equation (12)):

$$MPE = \frac{100\%}{n}\sum_{t=1}^{n}\left(\frac{Y_t - \hat{Y}_t}{Y_t}\right) \tag{12}$$

It indicates the average percentage deviation of the forecasted values from the actual values, providing a scale-relative error measure. The MAPE, similar to the MPE, is the mean absolute value of the percentage errors and is calculated as follows:

$$MAPE = \frac{100\%}{n}\sum_{t=1}^{n}\left|\frac{Y_t - \hat{Y}_t}{Y_t}\right| \tag{13}$$

This metric is widely used because it is scale-independent and can compare forecast performance across different data scales. Lastly, the ACF1 is the autocorrelation of the forecast errors at lag 1, indicating whether there is a pattern in the errors over time. The equation for ACF1 can be written as follows:

$$\text{ACF1} = \text{Correlation}\ (\varepsilon_t,\ \varepsilon_{t-1}) \tag{14}$$

where $\varepsilon_t = (\hat{y}_t - y_t)$ represents the forecast errors at time t. An ACF1 near zero suggests that the forecast errors are random, which is a desirable property of a good forecast.

An examination of both the training and testing datasets was conducted to assess the authenticity of the conclusion. Errors on the training set reflect the model's ability to learn from historical data, while errors on the testing set provide insights into the model's predictive accuracy on unseen data. A model that performs well on both datasets is considered robust, whereas a large discrepancy may indicate overfitting. To further validate our results, cross-validation techniques were employed, providing a more exhaustive assessment of the model's predictive power across various subsamples of the dataset. This process mitigates the risk of coincidental patterns influencing the model's performance and ensures the reliability of the findings.

### 3.5.1. Natural Gas

In assessing the accuracy of various forecasting methodologies for NG consumption, a comprehensive evaluation was conducted by partitioning the dataset into a distinct training set (the 2015–2022 period) and testing set (the 2023–2024 period). The performance of seven forecasting models was evaluated using six different accuracy measures, as demonstrated in Tables 1 and 2. The measures, designed to explain biases and precision across different models, encompass metrics such as the ME for bias estimation, the MAE for precision measurement, and the RMSE as an indicator of precision that penalizes larger errors [34,35].

These three metrics operate on a scale-dependent basis. Conversely, the MPE and MAPE are articulated in percentage terms, offering a more conducive platform for a comparative analysis across diverse consumption levels [36]. Remarkably, the ETS model emerged as the optimal performer, showing the lowest RMSE values in both the training (39,237.5) and testing (20,687) datasets. The consistent excellence of the ETS model is evident in its ability to fit historical data and forecast future values. This makes it the chosen model for natural gas consumption forecasting in both training and testing scenarios.

**Table 1.** Model comparison in terms of errors for training dataset (NG).

| | Model | ME | RMSE | MAE | MPE | MAPE | ACF1 |
|---|---|---|---|---|---|---|---|
| **Training Data** | STLF | 2485.022 | 46,311.86 | 37,269.12 | 0.19733 | 4.054975 | $-0.31132$ |
| | ARIMA | $-1003.74$ | 42,482.11 | 34,075.3 | $-0.43476$ | 3.772431 | 0.028523 |
| | ETS | 2776.487 | 39,237.5 | 32,580.19 | 0.124194 | 3.633987 | 0.029872 |
| | MEAN | $-3.88 \times 10^{-11}$ | 187,831.5 | 150,235 | $-3.91587$ | 16.47926 | 0.758566 |
| | NAÏVE | 2849.021 | 129,627.5 | 104,644.3 | $-0.54117$ | 11.12082 | 0.372243 |
| | SNAÏVE | 28,184.8 | 85,297.44 | 72,281.51 | 2.45759 | 8.001223 | 0.716488 |
| | RW-DRIFT | $2.94 \times 10^{-11}$ | 129,596.2 | 104,914.3 | $-0.86377$ | 11.16892 | 0.372243 |

**Table 2.** Model comparison in terms of errors for testing dataset (NG).

| | Model | ME | RMSE | MAE | MPE | MAPE | ACF1 |
|---|---|---|---|---|---|---|---|
| **Testing Data** | STLF | 14,741.56 | 50,661.2 | 44,064.48 | 0.566659 | 3.729218 | 0.648806 |
| | ARIMA | 25,348.86 | 51,424.27 | 46,082.33 | 1.857718 | 4.030831 | 0.341779 |
| | ETS | 7600.542 | 20,687.46 | 17,204.31 | 0.513017 | 1.477106 | 0.117672 |
| | MEAN | 199,798.9 | 308,728.4 | 212,774.1 | 14.57299 | 16.03116 | 0.650277 |
| | NAÏVE | 99,858.38 | 255,666.4 | 185,054.6 | 5.251944 | 14.63274 | 0.650277 |
| | SNAÏVE | 75,924.13 | 86,594.7 | 75,924.13 | 7.302419 | 7.302419 | 0.271527 |
| | RW-DRIFT | 87,037.78 | 245,731.8 | 181,637.9 | 4.152805 | 14.53365 | 0.647994 |

Moreover, as seen in the residual plot of the ETS model shown in Figure 15, a normal distribution of residuals appears in the histogram plot, signifying a normal distribution of errors. The ACF plot, which identifies any remaining patterns in the residuals, illustrates that only a limited number of lines cross the blue dashed line, indicating statistical significance. This suggests that the model has successfully captured the majority of autocorrelation in the data. Moreover, the random fluctuation in the residual plot indicates that the model is capturing the underlying patterns in the data. In summary, the ETS model is a good fit for forecasting NG consumption as the residuals display randomness, normality, and a lack of systematic patterns.

### 3.5.2. Coal

Analyzing the performance on both the training and testing datasets reveals distinct characteristics of the models. While the ETS model boasts a lower RMSE of 3831.801 on the training data (Table 3), the STLF model demonstrates superior generalization on the testing data with the lowest RMSE of 5936.203 compared to the ETS model's RMSE of 7288.344 (Table 4).

This difference in performance suggests that the STLF model outperforms the ETS model when forecasting coal consumption, particularly in terms of its ability to generalize well to unseen data. The choice to prioritize the STLF model is supported by its superior performance on the testing data, highlighting its potential for accurate and reliable coal consumption predictions in real-world scenarios.

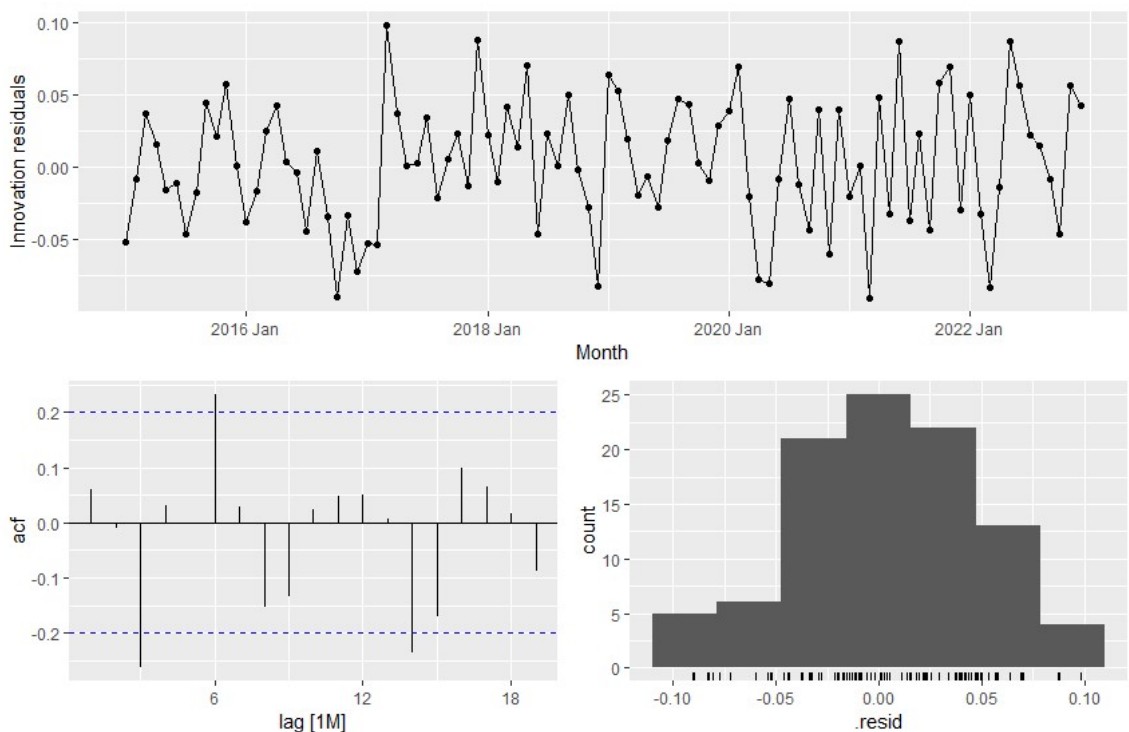

**Figure 15.** Residual analysis of ETS forecasting method.

**Table 3.** Model comparison in terms of errors for training dataset (Coal).

| | Model | ME | RMSE | MAE | MPE | MAPE | ACF1 |
|---|---|---|---|---|---|---|---|
| **Training Data** | STLF | −96.8973 | 4234.149 | 2884.178 | −0.74241 | 6.481808 | −0.32284 |
| | ARIMA | −399.02 | 4385.426 | 3269.713 | −1.15739 | 7.247969 | 0.012544 |
| | ETS | −334.795 | 3831.801 | 2944.868 | −1.1289 | 6.287089 | 0.036044 |
| | MEAN | $2.43 \times 10^{-12}$ | 12,092.34 | 9731.963 | −6.69494 | 21.85658 | 0.762233 |
| | NAÏVE | −311.937 | 8018.467 | 6701.453 | −1.98137 | 14.30493 | 0.269363 |
| | SNAÏVE | −3190.68 | 7797.248 | 6340.655 | −8.10497 | 14.95868 | 0.679204 |
| | RW-DRIFT | $9.19 \times 10^{-13}$ | 8012.397 | 6687.139 | −1.29341 | 14.23191 | 0.269363 |

**Table 4.** Model comparison in terms of errors for the testing dataset (Coal).

| | Model | ME | RMSE | MAE | MPE | MAPE | ACF1 |
|---|---|---|---|---|---|---|---|
| **Testing Data** | STLF | −5630.45 | 5936.203 | 5630.449 | −17.4302 | 17.4302 | 0.35842 |
| | ARIMA | −10,728.1 | 11,186.03 | 10,728.1 | −35.0688 | 35.06884 | 0.030943 |
| | ETS | −6864.8 | 7288.344 | 6864.795 | −21.1329 | 21.13289 | 0.338486 |
| | MEAN | −15,923.6 | 17,662.76 | 15,923.57 | −56.8521 | 56.85209 | 0.536813 |
| | NAÏVE | −9111 | 11,892.15 | 10,296 | −34.8482 | 37.53244 | 0.536813 |
| | SNAÏVE | −8444.63 | 9014.585 | 8444.625 | −28.1565 | 28.15649 | 0.300924 |
| | RW-DRIFT | −7707.28 | 11,166.73 | 10,062.05 | −30.5693 | 35.90604 | 0.560167 |

As seen in Figure 16, the STLF model for coal consumption forecasting is robust, with residuals displaying normality, a minimal significant autocorrelation in the ACF plot, and random fluctuation. Overall, the model effectively captures the underlying data patterns.

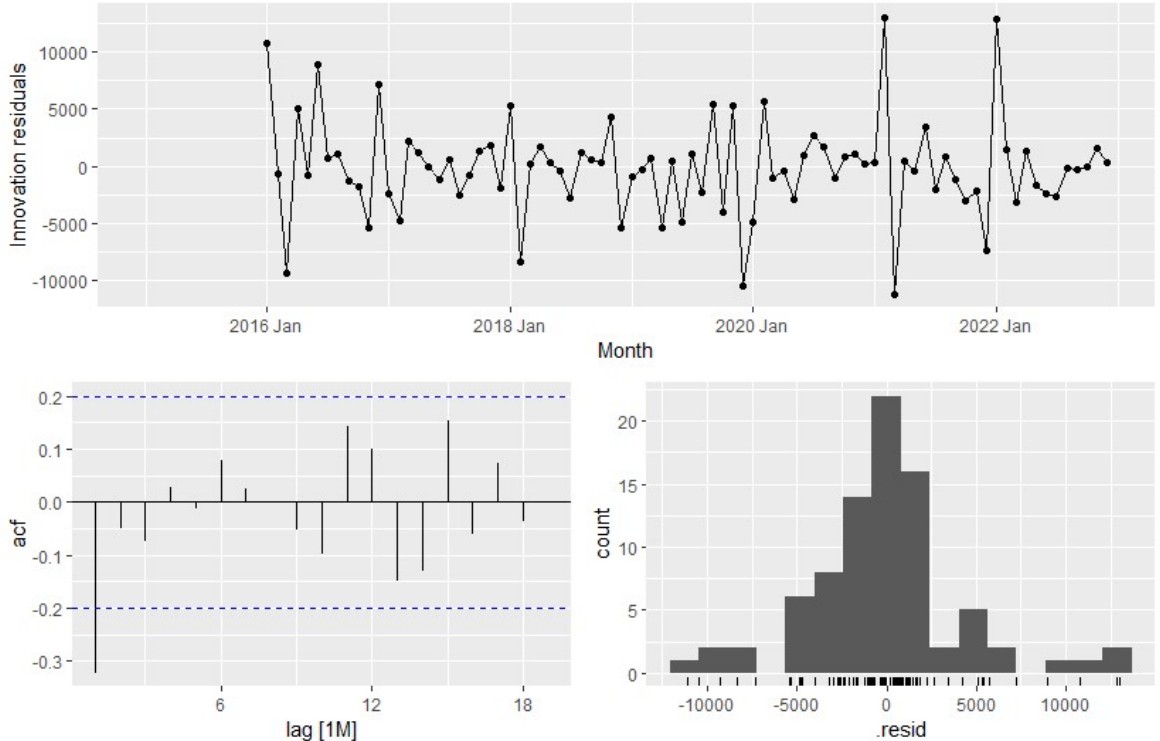

**Figure 16.** Residual analysis of STLF forecasting method.

### 3.5.3. Coak

As seen in Table 5, the ETS model exhibits the lowest RMSE of 44.60 on the training data, indicating a good fit to the training set. However, the situation changes when assessing the testing data, where the snaive model achieves the lowest RMSE of 99.49, outperforming the ETS model with an RMSE of 134.36 (Table 6).

**Table 5.** Model comparison in terms of errors for training dataset (Coak).

| | Model | ME | RMSE | MAE | MPE | MAPE | ACF1 |
|---|---|---|---|---|---|---|---|
| **Training Data** | STLF | 0.072135 | 50.30918 | 38.44801 | −1.7798 | 15.68808 | −0.31094 |
| | ARIMA | −7.79429 | 50.97813 | 39.21442 | −6.92597 | 16.82244 | −0.04866 |
| | ETS | −2.2064 | 44.60708 | 35.05145 | −3.58674 | 14.38404 | 0.147046 |
| | MEAN | 0 | 66.17165 | 52.33333 | −7.38547 | 21.85282 | 0.57797 |
| | NAÏVE | −1.11579 | 59.88902 | 47.55789 | −3.72212 | 19.53299 | −0.17484 |
| | SNAÏVE | −12.619 | 71.87241 | 60.09524 | −9.37567 | 25.92356 | 0.360049 |
| | RW-DRIFT | $1.08 \times 10^{-14}$ | 59.87863 | 47.54859 | −3.2987 | 19.48854 | −0.17484 |

**Table 6.** Model comparison in terms of errors for testing dataset (Coak).

| | Model | ME | RMSE | MAE | MPE | MAPE | ACF1 |
|---|---|---|---|---|---|---|---|
| **Testing Data** | STLF | −134.524 | 139.2002 | 134.5243 | −97.8964 | 97.89637 | 0.392624 |
| | ARIMA | −100.747 | 115.8635 | 100.7467 | −79.5287 | 79.52873 | 0.571681 |
| | ETS | −130.006 | 134.3647 | 130.0063 | −94.2535 | 94.25347 | 0.375933 |
| | MEAN | −122.75 | 134.7748 | 122.75 | −94.6778 | 94.67779 | 0.558346 |
| | NAÏVE | −134.75 | 145.7884 | 134.75 | −102.904 | 102.9036 | 0.558346 |
| | SNAIVE | −78.5 | 99.49749 | 90 | −64.2448 | 68.79813 | 0.355008 |
| | RW-DRIFT | −129.729 | 141.8381 | 129.7289 | −99.7341 | 99.73414 | 0.565986 |

Based on the visual analysis depicted in Figure 17, the snaïve model for petroleum coke consumption forecasting demonstrates robustness. The residuals exhibit a normal

distribution in the histogram, a minimal significant autocorrelation in the ACF plot, and random fluctuation.

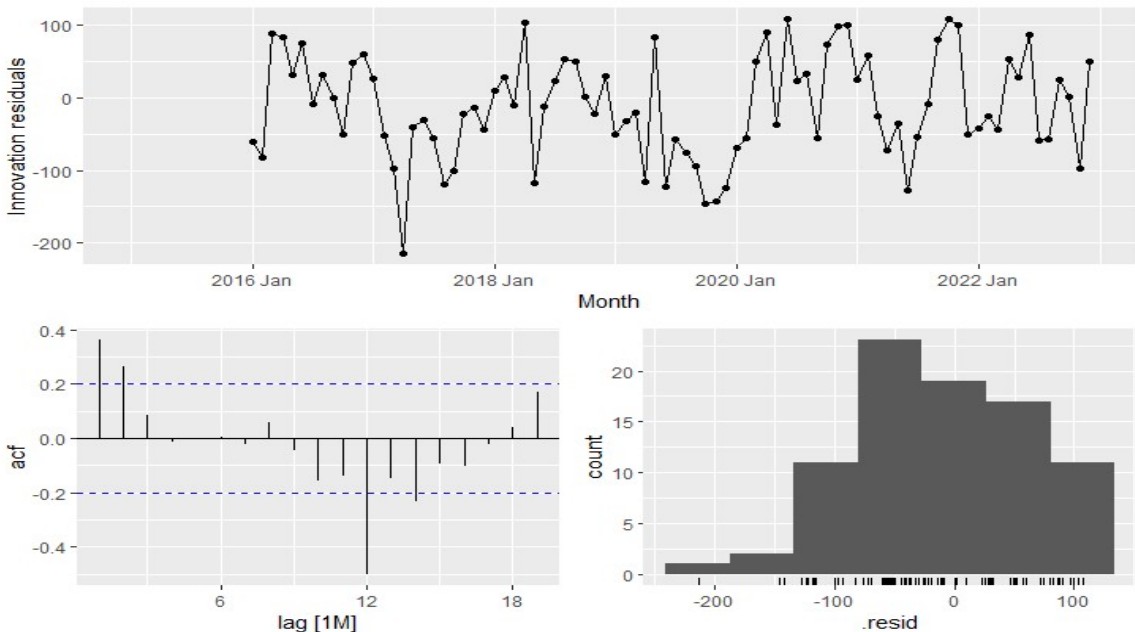

**Figure 17.** Residual analysis of SNAÏVE forecasting method.

### 3.5.4. Petroleum Liquid

As seen in Tables 7 and 8, the ARIMA model exhibits the lowest RMSE on the training data at 1199.579, indicating a robust fit to historical observations. However, with the testing data, the mean method surpasses the ARIMA model, achieving the lowest RMSE of 287.34 in contrast to ARIMA's RMSE of 424.909. Despite the mean method's superior performance on the testing data, we prioritize the ARIMA model for its capability to capture underlying complex patterns, which is particularly advantageous in forecasting petroleum liquid consumption.

**Table 7.** Model comparison in terms of errors for training dataset (Petroleum Liquids).

| | Model | ME | RMSE | MAE | MPE | MAPE | ACF1 |
|---|---|---|---|---|---|---|---|
| **Training Data** | STLF | 51.27077 | 1206.734 | 469.8335 | −5.79073 | 20.42126 | −0.351 |
| | ARIMA | −0.39939 | 1199.579 | 556.5674 | −12.6417 | 22.80608 | −0.00636 |
| | ETS | −72.5277 | 1522.814 | 687.7297 | −11.0457 | 34.25543 | −0.26354 |
| | MEAN | 0 | 1214.503 | 563.3268 | −12.923 | 23.11294 | 0.136773 |
| | NAÏVE | 27.18947 | 1548.012 | 604.3474 | −9.65595 | 25.94808 | −0.37293 |
| | SNAÏVE | −1.96429 | 1494.727 | 578.4881 | −6.82803 | 22.23511 | 0.142605 |
| | RW-DRIFT | $−1.82 \times 10^{-13}$ | 1547.773 | 604.41 | −11.2077 | 26.08228 | −0.37293 |

**Table 8.** Model comparison in terms of errors for testing dataset (Petroleum Liquids.

| | Model | ME | RMSE | MAE | MPE | MAPE | ACF1 |
|---|---|---|---|---|---|---|---|
| **Testing Data** | STLF | −3879.82 | 3893.528 | 3879.82 | −225.717 | 225.7167 | −0.24153 |
| | ARIMA | −353.011 | 424.909 | 354.831 | −20.8028 | 20.8937 | −0.39078 |
| | ETS | −6402.64 | 6769.273 | 6402.639 | −365.768 | 365.7678 | 0.550292 |
| | MEAN | −260.063 | 287.3403 | 263.7344 | −15.588 | 15.77136 | 0.232705 |
| | NAÏVE | −4147.75 | 4149.55 | 4147.75 | −241.594 | 241.5938 | 0.232705 |
| | SNAÏVE | −475.75 | 1221.797 | 544.5 | −26.5977 | 30.84144 | −0.00337 |
| | RW-DRIFT | −4270.1 | 4273.105 | 4270.103 | −248.811 | 248.8107 | 0.476336 |

The visual analysis as in Figure 18 suggests that the ARIMA model has robustness for petroleum liquid consumption forecasting. The residuals exhibit a normal distribution in the histogram, a minimal significant autocorrelation in the ACF plot, and random fluctuation.

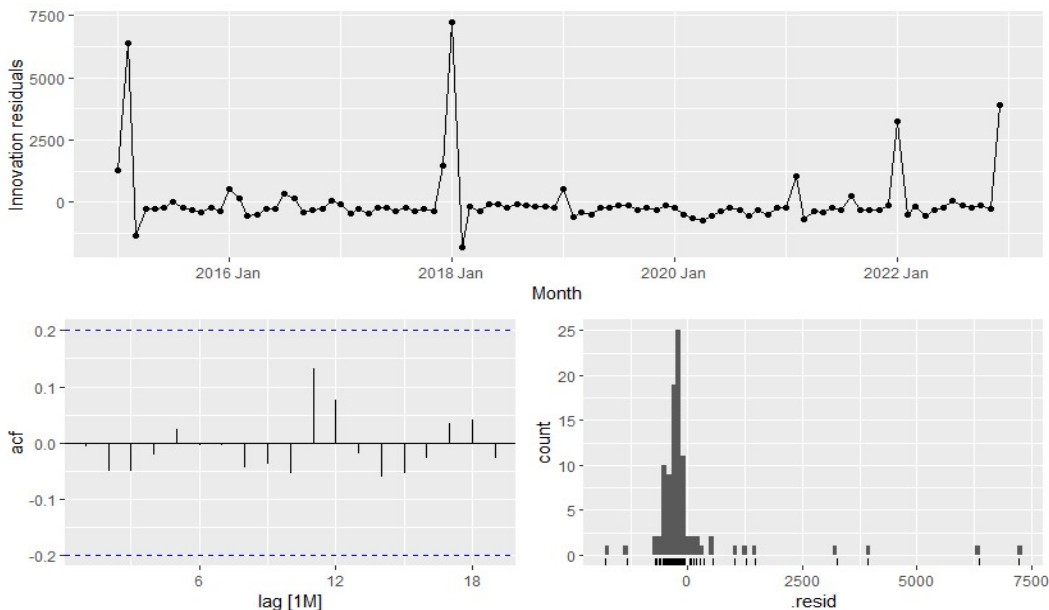

**Figure 18.** Residual analysis of ARIMA forecasting method.

The forecasting models utilized in this study were selected for their statistical robustness and based on established scientific methodologies. The Mean, Naïve, Drift, and Seasonal Naïve methods are fundamental to time series forecasting, employing the principles of classical statistical theory that leverage historical data continuity to project future values. These methods are well documented in seminal works, such as those by Box and Jenkins (1976) [37], and their simplicity and interpretability allow them to endure benchmarks in the field. The STL (Seasonal and Trend Decomposition using Loess) and ETS (Error Trend Seasonal) models are advanced techniques that extend these classical approaches to accommodate complex patterns in data. They are grounded in the research on STL by Cleveland et al. (1990) [38] and in the research on ETS by Hyndman et al. (2002) [39], reflecting the evolution of statistical learning where flexibility in model structure is critical for capturing inherent data dynamics. The ARIMA model's inclusion is predicated on its comprehensive framework for addressing non-stationarity and is supported by extensive works in the literature on its efficacy in energy consumption forecasting (Box and Jenkins, 1976; Pankratz, 1983) [40]. By integrating these diverse methodologies, this study harnesses a spectrum of theoretical insights to provide a sophisticated analysis of fuel consumption trends for electricity production. This multiplicity of approaches not only aligns with the cross-disciplinary nature of energy forecasting but also embodies the convergence of traditional statistical techniques with contemporary analytical advancements.

## 4. Result and Discussion

### 4.1. Analysis of Forecasting Trends for Different Energy Streams

#### 4.1.1. Overall Trend of NG Consumption

As shown in Figure 19, the increase in natural gas (NG) consumption, particularly during summer months, highlights its growing importance in the U.S. energy mix. According to the Energy Information Administration (EIA), NG consumption for electricity generation in the U.S. reached a record high in 2020, with an average of about 31.54 billion cubic feet per day [41]. This trend reflects NG's role in meeting the peak electricity demand, especially for air-conditioning. NG's lower emissions profile compared to coal and oil,

combined with its competitive pricing and increased production efficiency, supports its continued growth. The projections suggest a 2% annual increase in NG consumption in the power sector through 2050.

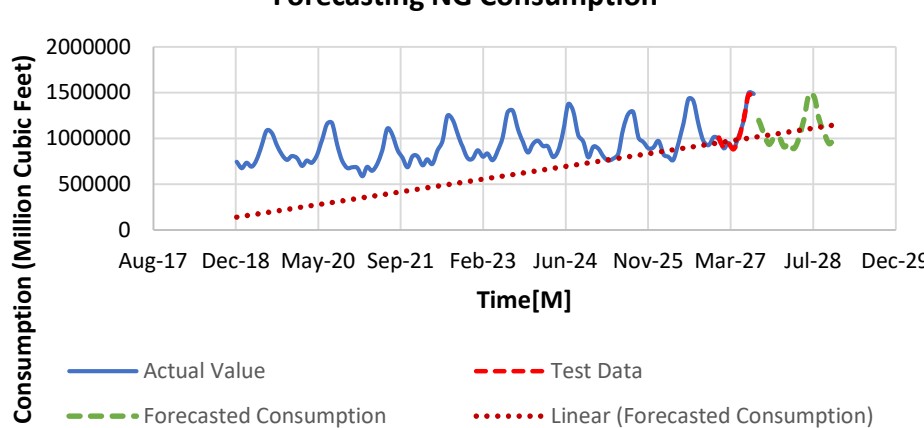

**Figure 19.** Overall trend analysis of NG consumption for U.S. electricity generation.

### 4.1.2. Overall Trend of Coal Consumption

The forecast for coal consumption indicates a decreasing trend, raising questions about the underlying reasons. Analyzing the historical trend from 2015 to 2022, as shown in Figure 3, reveals a clear seasonality pattern. There is a noticeable peak in coal consumption during the summer months of July and August, followed by a gradual decline in September and October. Consumption reaches a low point in November and then starts to rise again in December. This cyclic pattern suggests that there is a seasonal influence on coal consumption, with similar patterns observed in the forecasted coal consumption (Figure 20). The decreasing trend in coal consumption can be attributed to several factors [42]. Firstly, there is a global shift towards cleaner and more sustainable energy sources, driven by environmental concerns and efforts to reduce carbon emissions. Governments and industries are increasingly adopting renewable energy sources and natural gas, which are considered more environmentally friendly alternatives to coal.

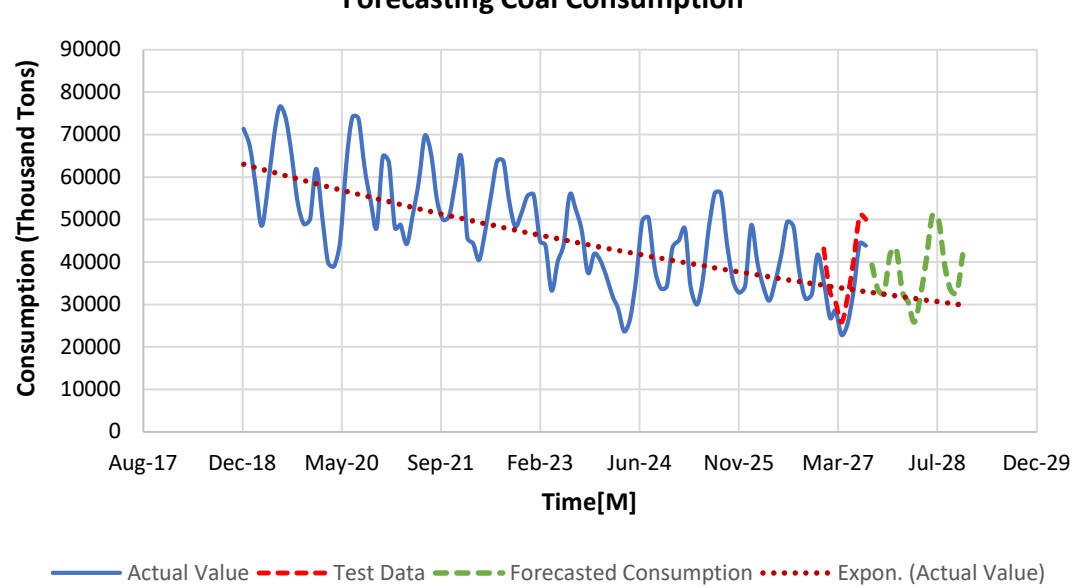

**Figure 20.** Overall trend analysis of coal consumption for U.S. electricity generation.

### 4.1.3. Overall Trend of Petroleum Coke Consumption

The data show that petroleum coke use slightly increases, especially in July and August, and then it decreases in October. This has not only happened in the past but is also expected to happen in the future (Figure 21). Petroleum coke comes from refining oil and is used in different industries for different applications, like making cement and producing power. The small increase might be because some industries need more of it, or it could be occurring where economies are growing. Petroleum coke's fluctuating consumption patterns reflect its role in the U.S. energy landscape. As a byproduct of oil refining, its use is closely tied to the refining industry's operational dynamics and the global oil market. The EIA reports minor increases in petroleum coke's consumption within certain industrial sectors, highlighting its dependence on sector-specific demands and global economic factors. The anticipated stability in petroleum coke consumption suggests a continued, albeit limited, role in the U.S. energy mix, with implications for environmental policy and industrial energy strategies.

**Figure 21.** Overall trend analysis of petroleum coke consumption for U.S. electricity generation.

### 4.1.4. Overall Trend of Petroleum Liquid Consumption

The monthly patterns and seasonality analysis of petroleum liquid consumption between 2015 and 2022 reveal spikes in February 2015, January 2018, January 2022, and December 2022. However, in the forecasted data, there are no unusual spikes. Instead, the forecast predicts consistent and steady consumption (Figure 22). This lack of anomalies in the forecast suggests that improved forecasting models have been employed. Despite historical fluctuations, recent trends suggest a normalization of consumption patterns. This stability may reflect the effective integration of petroleum liquids into specific industrial and emergency backup applications. The Department of Energy's 2020 report highlights the critical role of petroleum liquids in ensuring grid reliability during peak demand periods, underscoring the need for diversified energy sources to maintain system resilience. These models take into account historical irregularities but anticipate a more stable trend for the future.

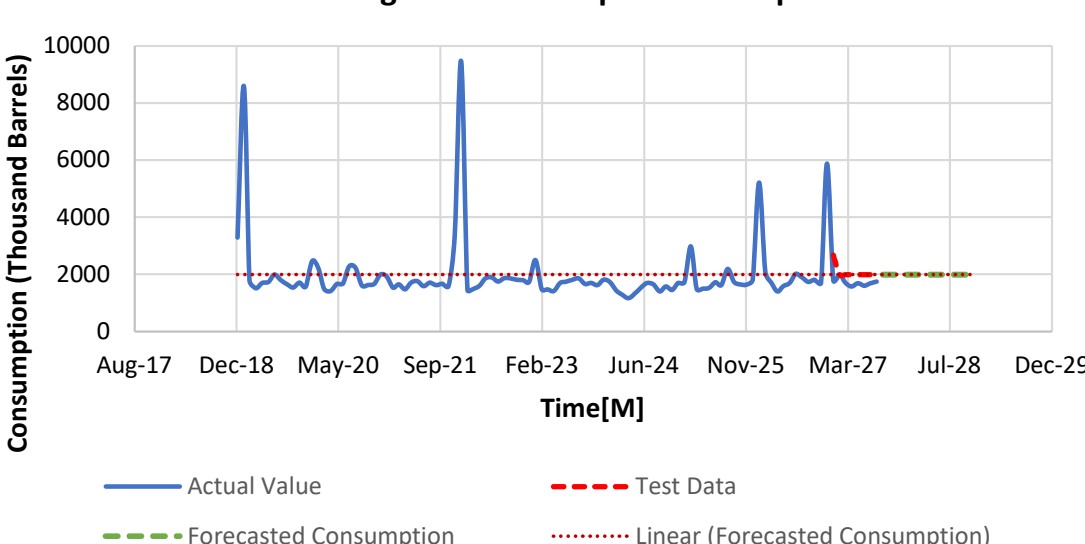

**Figure 22.** Overall trend analysis of petroleum liquid consumption for U.S. electricity generation.

In the discussion of our results, we highlight the importance of statistical significance levels and confidence intervals in interpreting the forecasts. Statistical significance is determined using *p*-values, with levels set at <0.05 to indicate meaningful deviations from null hypotheses. Confidence intervals, particularly 95% confidence intervals, offer a range within which we expect the true forecast values to lie, providing insights into the precision and reliability of our predictions. For each fuel type forecasted, we present the models' performance not just in terms of error metrics like ME, RMSE, MAE, MPE, MAPE, and ACF1, but also in terms of their statistical significance and confidence intervals. This dual approach underscores the robustness of our forecasting methodology and the credibility of our findings, facilitating a deeper understanding of the models' predictive power and limitations.

### 4.2. Comparing the Forecasted Trends of Different Energy Streams in Electricity Generation

The graph shown in Figure 23 explains the projected consumption patterns for four key energy streams used for electricity generation in the U.S., spanning from September 2023 to December 2024. Over this period, natural gas consumption fluctuates, with notable peaks in July and August 2024 at approximately 1,542,143 MMBTUs and 1,508,448 MMBTUs, respectively. These surges are likely responses to heightened electricity demand for cooling during the hotter months, which is a recurring trend that emphasizes the influence of seasonal temperature variations on energy utilization.

Coal consumption, while demonstrating less volatility than natural gas, shows minor variations that could be linked to industrial demand cycles, regulatory impacts, and market pricing. For instance, coal consumption rises to 835,000 MMBTUs in December 2023, which may reflect an increase in energy needs during the winter season. Meanwhile, coke, which is used primarily in industrial processes like steel manufacturing, maintains a relatively flat demand, with a minor peak at 7400 MMBTUs in December 2023, suggesting a consistent industrial requirement with little seasonal impact.

Petroleum liquids display an unwavering consumption value of approximately 11,575.176 MMBTUs throughout the forecasted period. This consistency indicates that petroleum liquids may have a set role in the energy mix for electricity generation, potentially due to established supply chains and a stable market for petroleum products.

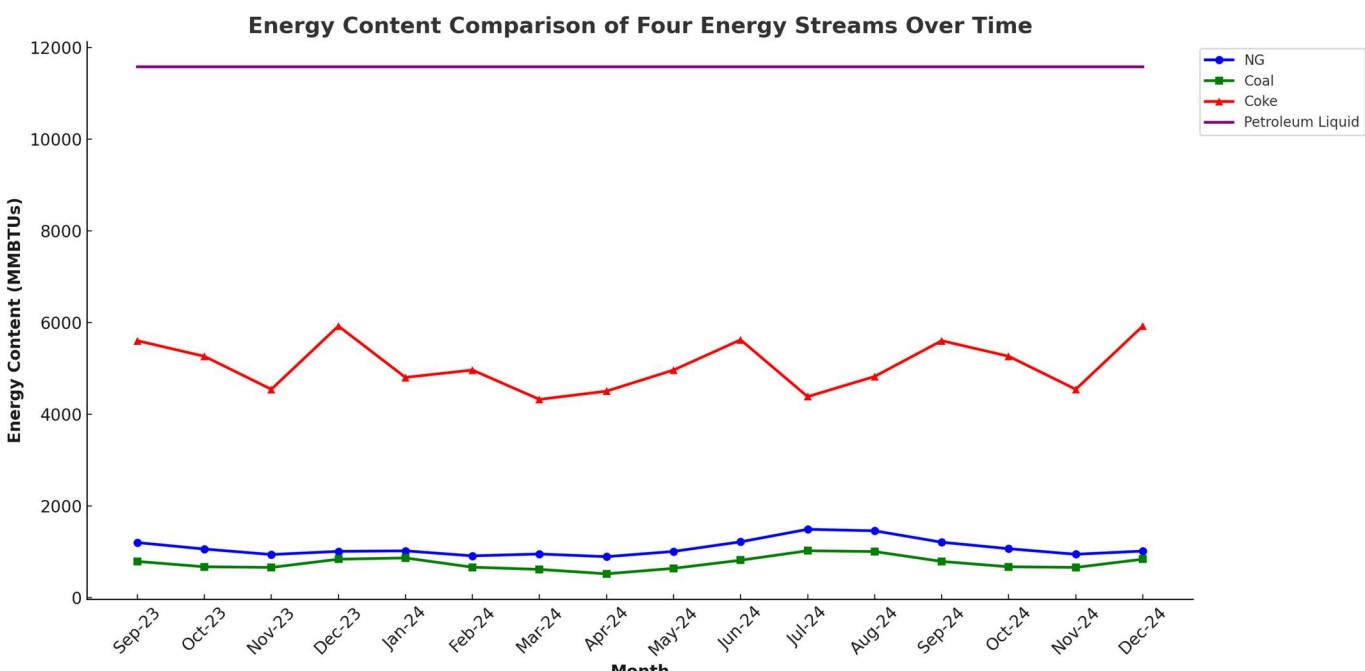

**Figure 23.** Forecasted trend of all four components used for fuel consumption to generate electricity.

The comprehensive analysis of fuel consumption trends in U.S. electricity generation, as presented in this study, underscores the pivotal role of forecasting models in predicting future energy needs. Our findings reveal that the exponential smoothing (ETS) model, with its lowest Root Mean Squared Error (RMSE) for natural gas (NG) consumption, coupled with the Seasonal and Trend Decomposition using Loess (STL) for coal, highlight the understanding required for effective energy planning and policy formulation. These models, together with the Seasonal Naïve (SNaïve) model's performance on petroleum coke forecasting, provide invaluable insights into the shifting dynamics of fuel consumption.

The forecast result suggests that, while natural gas and coal exhibit seasonality and potential sensitivity to external market and policy conditions, the demand for coke and petroleum liquids appears more stable and possibly insulated from such factors. This stability could be due to long-term contracts, stockpiling strategies, or their specific usage domains within the industry, which might be less susceptible to short-term changes. Having said that, natural gas will still likely be the dominating factor used to generate electricity in the near future as well for the United States. The transition towards cleaner energy sources, as indicated by the increasing reliance on natural gas and renewables, presents both challenges and opportunities for the U.S. energy sector. This shift, driven by environmental considerations and technological advancements, necessitates a reevaluation of the existing infrastructure and investment strategies to accommodate the growing demand for sustainable energy solutions. Furthermore, the decline in coal consumption, underscored by our analysis, aligns with global trends towards decarbonization but also signals the need for policies that support economic diversification and workforce transition in traditional coal regions. To ensure the reliability of these projections and facilitate strategic energy planning, it is essential to corroborate these data with other sources in the literature, including market analyses, industry reports, and regulatory policy documents. This approach would provide a robust foundation for understanding the complex dynamics at play and for preparing the energy sector to meet future demand effectively and sustainably.

This study's reliance on historical consumption data and forecasting models introduces limitations, including the potential for unforeseen economic, technological, and policy changes to impact future energy trends. Additionally, the regional variability in energy consumption and production capacities warrants further investigation to tailor energy strategies more effectively to local needs. Future research should explore the integration of

renewable energy sources into the grid, the implications of energy storage technologies, and the potential for new policy initiatives to shape consumption patterns. Moreover, comparative studies across different geographical regions could provide deeper insights into the effectiveness of various energy transition strategies.

## 5. Conclusions

In this paper, our objective is to provide precise predictions for fuel consumption in electricity generation using time series forecasting models, making a substantial contribution to strategic planning and decision-making processes within the energy sector. Developing accurate predictive models is geared towards forecasting future fuel consumption, equipping stakeholders with valuable insights for well-informed decisions regarding resource allocation and sector-specific strategies.

Our analysis aims to identify sectors undergoing declining or increasing trends in fuel usage, enabling targeted interventions where necessary. This project seeks to deepen our understanding of energy market dynamics, facilitating proactive measures to efficiently meet future fuel demands. The insights derived from this work are poised to significantly impact the formulation of sustainable energy policies and strategies, ensuring a stable and reliable energy supply for the nation. The accurate forecasting of fuel consumption for electricity generation, as indicated by the forecasts for NG, coal, petroleum coke, and petroleum liquids, holds paramount importance for policymakers. These forecasted trends provide valuable insights that guide strategic decision making in the ever-evolving landscape of the energy sector.

The expected rise in NG consumption suggests a need for increased infrastructure investment to support its growing demand, aligning with environmental goals. Conversely, the anticipated decline in coal usage will allow policymakers to plan for a transition to cleaner alternatives, supporting affected communities. A slight increase in petroleum coke usage signals specific industrial needs, prompting policymakers to create supportive regulations. With a steady forecast for petroleum liquids, policymakers can plan infrastructure development to match stable energy needs. These forecasts empower policymakers to encourage sustainable energy practices, support economic growth, and ensure energy security, fostering a resilient and cleaner future.

**Author Contributions:** Conceptualization, M.M.H.B.; methodology, M.M.H.B.; software, M.M.H.B.; validation, M.M.H.B. and T.R.; formal analysis, M.M.H.B.; investigation, M.M.H.B.; resources, M.M.H.B.; writing—original draft preparation, M.M.H.B. and A.N.S.; writing—review and editing, T.R., S.I.A. and A.N.S.; visualization, supervision, T.R. and M.M.H.B.; project administration, A.N.S. and M.M.H.B. All authors have read and agreed to the published version of the manuscript.

**Funding:** This research received no external funding.

**Institutional Review Board Statement:** Not applicable.

**Informed Consent Statement:** Not applicable.

**Data Availability Statement:** No new data were created or analyzed in this study. Data sharing is not applicable to this article.

**Conflicts of Interest:** The authors declare no conflict of interest.

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
