# Peer review of "Fueling the Future: A Comprehensive Analysis and Forecast of Fuel Consumption Trends in U.S. Electricity Generation"

_sustainability, doi:10.3390/su16062388_

Round 1
Reviewer 1 Report
Comments and Suggestions for Authors
The article addresses the issue of fuel consumption for electricity generation in the US. The data analyzed are those provided by the EIA US monthly. Coal, Petroleum Liquids, Petroleum Cook and Natural Gas are the fuels included. The objective is to predict fuel consumption trends based on the analyzed data. For them, they use various time series prediction models.
Some observations regarding the order of the article are that the end of the introduction does not mention the outline of the work and that it is joined with the first analysis of the information in a kind of subsection without numbering that could well be the beginning of the next section of methodology. Methodology section 2 also includes data analysis, so its title should be improved.
In line 100, Fig 1 is generally mentioned, but in lines 101 to 121, the specific citation of each subplot needs to be added where it corresponds; by the way, the caption of Fig 1d) needs to be corrected.
During the paragraph from lines 109 to 121, some affirmation phrases are mentioned without citation or justification; that is the case in lines 109-110, line 119, and line 121.
In line 148, does the citation of Fig 3 refer to Fig 3a)? Discussion of Fig 3a only includes the second half of the year. The first half must be discussed also. In section 2.1.2, a discussion of Fig 3b) needs to be included.
In section 2.2, it is desirable to include a mathematical formulation of ACF as PACF does and also describe how AR and MA components are obtained from them. Also, in this section, the author shall mention what they mean by confidence interval and why they take a 95% value. Why is lag 36 acceptable for the study?
The arrangement of graphs in Figures 6, 7, 8, and 9 needs subplot identification, for example, Fig 6a) and b). This helps refer to each one appropriately in their corresponding paragraph for a better understanding.
The description in lines 243-244 does not correspond to the observed behavior in Fig 7.
The legend in line 267 needs to be corrected. It must be Petroleum Liquids.
Part of the discussion presented in section 2.3.1 about the Forecasting Methods is general, not only for NG consumption. That must be extracted and included in the introduction part of section 2.3.
Bibliographic citations need to be included for all the methods mentioned in lines 282, 283, and 289-294.
Each sub-subsection for 2.3.1 needs to be referred to the corresponding subplot of Fig 10. An equivalent argument applies to the other subsections 2.3.
In section 2.4, no description of accuracy measures is presented. Also, no bibliographic citations are included for them. Thus, it is unclear why they are adequate for these analyses.
From my point of view, the rest of the document, including discussion and concluding remarks, is meaningless if the previous recommendations are not clarified previously.
Reviewer 2 Report
Comments and Suggestions for Authors
I have had the privilege of reviewing the manuscript entitled "Fueling the Future: A Comprehensive Analysis and Forecast of Fuel Consumption Trends in U.S. Electricity Generation" submitted by [Author Names]. The paper presents an in-depth analysis of an important topic in the field of energy and electricity generation. Overall, the manuscript shows promise but requires significant revisions to meet the standards of this journal.
Below are my major comments and suggestions for improvement:
I. The paper lacks clear and structured organization. It would greatly benefit from a more structured approach, including subsections that clearly delineate different aspects of the analysis (e.g., methodology, data sources, results, discussion). This would enhance the readability and comprehension of the paper.
II. The paper does not provide a detailed description of the data sources and the methodology used for the analysis. It is crucial to provide readers with a comprehensive understanding of the data collection process and the statistical techniques employed to ensure the validity of the results. Without this information, it is difficult to assess the robustness of the findings.
III. The discussion section of the paper is quite limited and lacks depth. The authors should provide a more detailed interpretation of the results, discussing the implications of their findings on the future of electricity generation in the U.S. Additionally, the authors should consider discussing potential limitations in their analysis and suggest avenues for future research.
IV. The paper relies heavily on textual descriptions of data trends, and it would benefit from the inclusion of well-designed figures and visualizations to illustrate key findings. Clear and informative charts, graphs, and tables can help convey complex information more effectively.
V. The manuscript requires a more comprehensive and up-to-date reference list. Several recent studies and key references in the field of electricity generation and fuel consumption trends appear to be missing. Authors should ensure that all relevant literature is cited appropriately.
VI. The statistical analysis in the paper needs further elaboration and explanation. Readers should be able to understand the statistical tests and models used. The use of appropriate statistical significance levels and confidence intervals should also be addressed.
VII. The manuscript contains numerous grammatical errors, awkward phrasings, and instances of unclear writing. The authors should carefully proofread and edit the manuscript to improve the overall quality of the writing.
Minor comments:
I. The resolution of all figures needs to be improved.
II. There is no clear mention of the statistical models used in prediction. The mathematical basis of the models must be carefully developed. It should be noted how to use it. Also, the correct programming code for this must be placed in the appendix.
III. For example, the results in the first and second tables: How were these results obtained and how can they be interpreted? How can we verify its authenticity?
IV.
In summary, this manuscript addresses an important topic but requires substantial revisions to improve its clarity, organization, and depth of analysis. I recommend that the authors address these major comments before resubmitting the paper for further consideration.
Comments on the Quality of English LanguageExtensive editing of English language required
Reviewer 3 Report
Comments and Suggestions for Authors
The article is written on a very relevant topic today. The authors clearly defined the purpose of the study: to forecast the trend of fuel consumption for electricity production. The article is logically structured, but the scientific schools and their approaches to building a predictive model are not clearly defined in the methodology. In the study, the methods of comparison (average, naive, drift and seasonally naive), time series forecasting models, seasonal decomposition and trend decomposition using Loess (STL), etc. are clearly defined. The root mean square error (RMSE) was used to determine the most effective model.
Another minor complaint is the insufficient number of references. There are also few or no comparisons and references to similar studies
Round 2
Reviewer 1 Report
Comments and Suggestions for Authors
The authors responded to almost all recommendations; minor changes are required. That is the case for figures 6 to 9, where titles are repeated. Also, they must number all the equations.
